# Self-Supervised Diffusion Models for Electron-Aware Molecular Representation Learning

**Gyoung S. Na**
KRICT, Republic of Korea
ngs0@krict.re.kr

**Chanyoung Park**
KAIST, Republic of Korea
cy.park@kaist.ac.kr

## Abstract

Physical properties derived from electronic distributions are essential information that determines molecular properties. However, the electron-level information is not accessible in most real-world complex molecules due to the extensive computational costs of determining uncertain electronic distributions. For this reason, existing methods for molecular property prediction have remained in regression models on simplified atom-level molecular descriptors, such as atomic structures and fingerprints. This paper proposes an efficient knowledge transfer method for electron-aware molecular representation learning. To this end, we devised a self-supervised diffusion method that estimates the electron-level information of real-world complex molecules without expensive quantum mechanical calculations. The proposed method achieved state-of-the-art prediction accuracy in the tasks of predicting molecular properties on extensive real-world molecular datasets.

## 1 Introduction

Machine learning has been widely studied as an efficient data-driven method for predicting the physical and chemical properties of molecules (Wigh et al., 2022). In particular, graph neural networks (GNNs) (Kipf & Welling, 2017) achieved numerous successes in various molecular representation learning tasks (Bilodeau et al., 2022; Duval et al., 2023). In GNNs, an atom-level molecular structure is defined as a graph $G = (\mathcal{V}, \mathcal{U}, \mathbf{A}, \mathbf{X}, \mathbf{R})$, where $\mathcal{V}$ is a set of nodes (i.e., atoms), $\mathcal{U}$ is a set of edges (i.e., chemical bonds), $\mathbf{A} \in \{0, 1\}^{|\mathcal{V}| \times |\mathcal{V}|}$ is an adjacency matrix, $\mathbf{X} \in \mathbb{R}^{|\mathcal{V}| \times d}$ is a $d$-dimensional node-feature matrix, and $\mathbf{R} \in \mathbb{R}^{|\mathcal{U}| \times r}$ is an $r$-dimensional edge-feature matrix (Wieder et al., 2020).

In addition to traditional GNNs, various methods have been proposed to learn informative molecular representations from different approaches, such as fragmentation-based learning (Zhang et al., 2021; Kim et al., 2023), domain knowledge integration (Wang et al., 2022), and hierarchical representation learning (Zang et al., 2023). However, existing methods aimed to learn molecular representations from *atom-level* molecular descriptors, while overlooking a physical principle that molecular properties are essentially derived from *electron-level* information, such as electronic distributions and related electronic energies, beyond the atom-level information (Parr & Yang, 1995; Engel & Dreizler, 2011). Therefore, the representation capabilities of the existing methods for molecular representation learning are inherently limited, even though they were sophisticatedly designed to capture latent information from the atom-level molecular descriptors.

In physical science, quantum mechanical calculations have been used as a de facto standard to calculate the electron-level molecular information (Parr & Yang, 1995), such as molecular orbital and atomization energy. However, as these methods suffer from cubic or greater time complexities with respect to the number of electrons in a molecule (Engel & Dreizler, 2011; Dawson et al., 2022), electron-level information about real-world complex and large molecules are usually not accessible in chemical applications. Although there is an efficient solution that uses sophisticatedly designed 3D-GNNs with electron-level information generated by force-field-based and semi-empirical calculations (FFSECs) (Riniker & Landrum, 2015), the effectiveness of this straightforward solution is questionable due to the low calculation accuracy of the FFSEC methods. To corroborate our argument, we empirically evaluated the effectiveness of 3D-GNNs with FFSEC on well-known real-world molecular datasets: Lipop (Wu et al., 2018), ESOL (Delaney, 2004), and IGC50 (Wu & Wei, 2018) datasets. Fig. 1 shows

that the experimental evaluations did not demonstrate significant improvement by the 3D-GNNs with FFSEC, denoted by PhysChem (Yang et al., 2021), M3GNet (Chen & Ong, 2022), and FAENet (Duval et al., 2023), while a simple 2D-GNN called AttFP (Xiong et al., 2019) rather showed better prediction accuracy. We conjecture two major reasons for such results: 1) The calculation errors of FFSEC can be propagated to the 3D

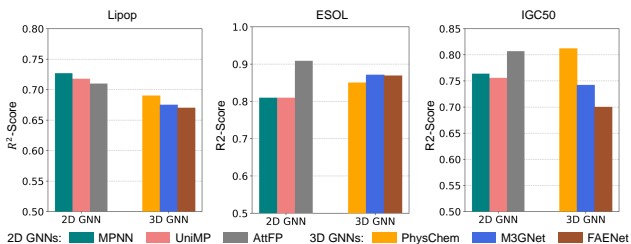

Figure 1: $R^2$-scores (Nagelkerke et al., 1991) of 3D-GNNs with FFSEC on real-world molecular datasets.

GNN models, which in turn degrades the prediction accuracy. 2) Complex 3D-GNNs are not effective in representation learning on large molecules due to the overfitting problem (Li et al., 2023a).

In this paper, we propose decomposition-supervised electron-level information diffusion (DELID) for an *electron-aware* molecular representation learning. The main challenge is that the electron-level information $s_0$ is usually unknown for a real-world molecule $G$. To this end, we propose a self-supervised diffusion model for estimating the unknown latent representation $s_0$ without its ground truth being accessible in the training process. As shown in Fig. 2a, DELID consists of two diffusion models. The diffusion model on $G$ aims to estimate the original molecule $G$ from the noise $G_T$, while the diffusion model on $s$ aims to estimate the unknown complete electron-level information $s_0$ of $G$ from the noise $s_T$. The main idea of DELID is to consider $G_T$ as decomposed substructures of $G$, and $s_T$ as their electron-level information, where $G_T$ and $s_T$ can be easily obtained from molecular decomposition algorithms (Liu et al., 2017), and public chemical databases (Ramakrishnan et al., 2014), respectively, without expensive quantum mechanical calculations. As illustrated in Fig. 2b, DELID employs the transition probability $p(G_{t-1}|G_t; G_0)$ of the diffusion process on $G$ as a self-supervision to learn the diffusion process from readily accessible $s_T$ to unknown $s_0$. In Section 3.2, we will mathematically show that the diffusion model on $s$ can be optimized by minimizing the KL divergence between $p(G_{t-1}|s_t; G_0)$ and $p(G_{t-1}|G_t; G_0)$.

In our experiments, we focus on evaluating the prediction capabilities of the machine learning methods on biased and relatively small *experimental datasets* rather than *simulated datasets* (e.g., QM9 dataset (Ramakrishnan et al., 2014)). Although the simulated datasets are useful for analyzing rough statistics on small molecules, they are not appropriate to evaluate the prediction capabilities of the machine learning methods on real-world molecular physics due to the following two reasons: 1) The simulated datasets do not contain complex and large molecules due to the large time complexity of the quantum mechanical calculations. 2) The simulated datasets do not sufficiently reflect the quantum mechanical uncertainty in real-world molecules (Sim et al., 2018). For these reasons, we used experimentally collected molecular datasets from physicochemistry, toxicity, pharmacokinetics, and optical applications to evaluate the practical potential of DELID. For all benchmark molecular datasets, DELID achieved state-of-the-art performance in predicting experimentally observed properties of real-world complex molecules. The contributions can be summarized as:

- A novel method called DELID for learning electron-aware molecular representations beyond atom-level molecular representations without expensive quantum mechanical calculations.

- A self-supervised diffusion mechanism to estimate the unknown electron-level information.

- The state-of-the-art prediction accuracy of DELID on extensive real-world molecular datasets containing experimentally collected complex molecules and their properties.

## 2 RELATED WORK

### 2.1 GRAPH NEURAL NETWORKS ON MOLECULES

2D-GNNs for molecular representation learning on the 2D molecular structures have been widely studied in chemical science due to their practicality and efficiency. SchNet (Schütt et al., 2017) and MEGNet (Chen et al., 2019) are graph convolutional neural networks for molecular representation learning on quantum mechanical principles in chemical bonds and local atomic substructures. They employed an atom-wise representation to learn geometric information of the molecules. MPNN (Gilmer et al., 2017) is a message-padding neural network that captures the quantum mechanics

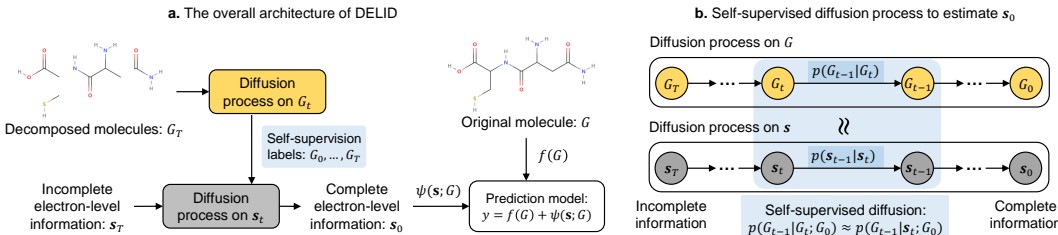

Figure 2: **a.** The overall architecture of DELID, and **b.** the self-supervised diffusion process to estimate the complete electron-level information $\mathbf{s}_0$ that represents the ground truth $\mathbf{s}$ of $G$. In the self-supervised diffusion process, $G_T$, $G_0$, and $\mathbf{s}_T$ are given data, whereas $\mathbf{s}_0$ is unknown data.

between the atoms. Directed MPNN (D-MPNN) is an extension of MPNN for molecular representation learning based on a directed message passing scheme (Yang et al., 2019). In addition to the general-purpose GNNs, AttFP (Xiong et al., 2019) was proposed to predict the physical and chemical properties of the molecules for drug discovery.

3D-GNNs that utilize the 3D molecular structures in representation learning have been devised for more accurate molecular representation learning. DimeNet++ (Gasteiger et al., 2020), PhysChem (Yang et al., 2021), and M3GNet (Chen & Ong, 2022), and FAENet (Duval et al., 2023) were proposed to learn local and global 3D geometry of the molecules. ConAN (Nguyen et al., 2024) used the 3D molecular structures of possible conformers of the input molecule in molecular representation learning. In addition to these methods, PaiNN (Schütt et al., 2021), GemNet (Gasteiger et al., 2021), Equiformer (Liao & Smidt, 2022), MolKGNN (Liu et al., 2023), and ViSNet (Wang et al., 2024) were proposed for molecular property prediction on the 3D structures. However, despite their state-of-the-art performances on several benchmark datasets on calculated molecular structures, their applicability is significantly limited in real-world molecular science because calculating an accurate 3D molecular geometry is not feasible in most real-world complex molecules due to the uncertainty of the electronic structures and the large computational costs to calculate them (Krivanek et al., 2010; Suenaga & Koshino, 2010; Schuch & Verstraete, 2009). Although ConAN showed state-of-the-art prediction accuracy with an efficient geometry calculation, its practicality also limited because we should generate all conformers of the input molecules for each inference process.

## 2.2 KNOWLEDGE TRANSFER METHODS FOR MOLECULAR REPRESENTATION LEARNING

Machine learning methods usually suffer from the lack of training data and informative features in chemical applications because time-consuming and labor-intensive chemical experiments are required to collect experimentally generated data (Gromski et al., 2019; Shen et al., 2021). To overcome the lack of experimental data, transfer learning to exploit simulated molecular data has been widely studied in physics and chemistry (Jha et al., 2019; Cai et al., 2020; Zaverkin et al., 2023; Dou et al., 2023b). However, the practicality of the transfer learning methods on simulated molecular data is still limited because the source calculation databases are not able to cover the majority of large molecules in real-world chemical experiments due to the cubic or greater time complexities of the calculation methods with respect to the number of electrons in the molecules (Engel & Dreizler, 2011; Schuch & Verstraete, 2009). In addition to transfer learning, various molecular representation learning methods have been proposed to exploit fragmented information on molecular representation learning (Zhang et al., 2021; Wang et al., 2022; Yu & Gao, 2022; Chen et al., 2022; Kim et al., 2023; Feng et al., 2023). However, the representation capabilities of the existing molecular representation learning methods are essentially limited to the atom-level molecular representations because they did not consider how to estimate the electron-level information and how to utilize it.

## 2.3 DIFFUSION MODELS

Diffusion models aim to learn a stochastic process that approximates the probability distribution of a given dataset (Kingma et al., 2021; Song et al., 2020). The diffusion models have achieved remarkable successes in learning physical systems consisting of complex and long-step stochastic processes over conventional generative models (Zeni et al., 2023; Yuan et al., 2023; Wu & Li, 2023). The reverse process of the diffusion models, which restore the data from noise, can be used as a generative model to generate new data (Ho et al., 2022; Vignac et al., 2022). In addition to the conventional diffusion models, conditional diffusion models were devised to generate new data of desired properties (Tashiro et al., 2021; Zhang et al., 2023; Zbinden et al., 2023). Since the electron-level information is

usually derived by complex and long-step physical processes (Parr & Yang, 1995; Hollingsworth & Dror, 2018), DELID employs the variational diffusion models (Huang et al., 2021) rather than shallow generative models, such as variational autoencoder (Kingma, 2013). However, as the existing diffusion models essentially require the ground truth data to learn the forward and reverse processes, they are not applicable to our problem where the electron-level information is not available.

## 3 PROPOSED METHOD

**Vanilla Diffusion Models.** The diffusion models essentially aim to learn the reverse process $p(\mathbf{s}_{t-1}|\mathbf{s}_t)$ to restore the original data $\mathbf{s}_0$ from noise $\mathbf{s}_T$ (Ho et al., 2022). The diffusion models are usually optimized by maximizing the following probability (Kingma et al., 2021):

$$\log p(\mathbf{s}) = \log E_{q(\mathbf{s}_{1:T}|\mathbf{s}_0)} \left[ \frac{p(\mathbf{s}_T) \prod_{t=1}^{T} p(\mathbf{s}_{t-1}|\mathbf{s}_t)}{\prod_{t=1}^{T} q(\mathbf{s}_t|\mathbf{s}_{t-1})} \right]. \tag{1}$$

A fundamental assumption of the diffusion models is that $\mathbf{s}_0$ to learn the diffusion processes is given in the training dataset (Kingma et al., 2021; Graham et al., 2023). However, since $\mathbf{s}_0$ represents the electron-level information of a real-world molecule, it is assumed to be unavailable in our task.

**Challenges in the Diffusion Processes of DELID.** In our regression setting, the objective function of the prediction models is given by:

$$\log p(y, \mathbf{s}, G) = \underbrace{\log p(y|\mathbf{s}, G)}_{\text{Section 3.1}} + \underbrace{\log p(\mathbf{s}|G)}_{\text{Section 3.2.2}} + \underbrace{\log p(G)}_{\text{Section 3.2.1}}. \tag{2}$$

where $y$ is the target molecular property corresponding to the atom-level molecular descriptor $G$, and $\mathbf{s}$ is hidden electron-level representation about $G$. However, calculating $p(\mathbf{s}|G)$ based on the vanilla diffusion models is not feasible because the ground truth $\mathbf{s}$ is unknown in our task. Hence, our proposed DELID adopts a self-supervised diffusion process for learning the distribution of $\mathbf{s}_0$, even though $\mathbf{s}_0$ is not given in the training process, which will be described in the following subsections. We will also mathematically show how we can approximate $p(\mathbf{s}|G)$ based on the diffusion process on $p(G)$ under some mild conditions on a set of substructures $G_T$ of the original molecule $G$.

### 3.1 DELID: DECOMPOSITION-SUPERVISED ELECTRON-LEVEL INFORMATION DIFFUSION

Experimental observations on the atomic systems contain measurement noise originated from the uncertainty of the electronic distributions (Robertson, 1929; Najm et al., 2009). In physical science, quantum mechanical calculations have been widely used to quantify the uncertainty of the electronic distributions (Engel & Dreizler, 2011; Parr & Yang, 1995). Following the convention in physical science, DELID employs the electron-level information calculated by the quantum mechanical calculations as supplementary information to correct the measurement noise as:

$$y = f(G) + \psi(\mathbf{s}; G), \tag{3}$$

where $y$ is the target property of a molecule $G$, $f$ is a structure encoder for the atom-level molecular descriptor $G$, and $\psi$ is a function to estimate quantum mechanical noise from the electron-level information $\mathbf{s}$, which is calculated by the quantum mechanical calculations.

To predict the target molecular property $y$ based on Eq. (3), DELID implements $f$ as a deterministic function based on 2D-GNNs and $\psi$ as a stochastic function derived from parameterized normal distribution $\mathcal{N}(\mu_y, \sigma_y^2)$, where $\mu_y = f_{y,\mu}(\mathbf{s}; G)$ and $\sigma_y = f_{y,\sigma}(\mathbf{s}; G)$ are parameterized mean and standard deviation, respectively. However, since $\mathbf{s}$ is not accessible in real-world complex and large molecules, we devised the self-supervised diffusion process to estimate unknown $\mathbf{s}$ from a given atom-level molecular descriptor $G$. In the following sections, we will formally define the self-supervised diffusion process to estimate unknown $\mathbf{s}$ for $\psi(\mathbf{s}; G)$.

### 3.2 SELF-SUPERVISED DIFFUSION PROCESSES

#### 3.2.1 DIFFUSION PROCESS ON MOLECULAR GRAPHS

The purpose of the self-supervised diffusion of DELID is to learn the electron-aware representation $\mathbf{s}$ without labeled data for training $\mathbf{s}$. To this end, we define a diffusion process of the molecule $G$ that will be used to guide the diffusion process on the unknown $\mathbf{s}$. The lower bound of $\log p(G)$ is

given by the variational diffusion model (Kingma et al., 2021) as:

$$\log p(G) \geq E_{q(G_1|G_0)}\left[\log p(G_0|G_1)\right] - D_{\text{KL}}(q(G_T|G_0)||p(G_T))$$
$$- \sum_{t=2}^{T} E_{q(G_t|G_0)}\left[D_{\text{KL}}(q(G_{t-1}|G_t,G_0)||p(G_{t-1}|G_t))\right], \quad (4)$$

where $D_{\text{KL}}(\cdot||\cdot)$ is the KL divergence, $G_0$ is the latent embedding that follows the distribution of the input atom-level molecular graph $G$, and $G_T$ is the noised data of $G$. Note that the diffusion process on $G$ does not guarantee $G_0 \approx G$ in our problem setting because the entire model is finally trained to maximize $\log p(y, \mathbf{s}, G)$ instead of $\log p(\mathbf{s}, G)$, as shown in Eq. (2).

The main difference between the vanilla diffusion models and our proposed diffusion process lies in the assumption on $G_T$, i.e., vanilla diffusion models assume $G_T$ as a random noise drawn, whereas DELID defines $G_T$ as atom-level decomposed substructures of $G$. Formally, in DELID, $G_T = (\mathcal{V}_T, \mathcal{U}_T, \mathbf{A}_T, \mathbf{X}_T, \mathbf{R}_T)$ is defined as a graph of graphs, where $\mathcal{V}_T = \{\mathcal{F}_1, \mathcal{F}_2, ..., \mathcal{F}_K\}$ is a set of decomposed substructures of $G$, $\mathcal{U}_T = \{\mathcal{E}_1, \mathcal{E}_2, ..., \mathcal{E}_K\}$ is a set of edges within each substructure, $\mathbf{A}_T$ is an adjacency matrix representing $\mathcal{U}_T$, $\mathbf{X}_T$ is the atom-feature matrix of the atoms in each substructure, $\mathbf{R}_T$ is the edge-feature matrix of $\mathcal{U}_T$. The diffusion probability on $G_T$ is parameterized by GNN to consider the interactions between the substructures and their electron-level information.

**Definition 1.** *Complete graph decomposition. A graph decomposition is complete for $G$ if the decomposed substructures of $G$, denoted by $\mathcal{F}_1, \mathcal{F}_2, ..., \mathcal{F}_K$, satisfy $\mathcal{F}_1 \cup \mathcal{F}_2 \cup \cdots \cup \mathcal{F}_K = G$ and $\mathcal{F}_1 \cap \mathcal{F}_2 \cap \cdots \cap \mathcal{F}_K = \emptyset$ where $K$ is the number of decomposed substructures.*

DELID assumes that $G_T$ is decomposed substructures generated through a complete graph decomposition defined by Definition 1. The essential property of the complete graph decomposition is that the nodes in $G$ are preserved through the overall graph decomposition process, i.e., $\mathbf{X}_T$ is preserved through the overall graph decomposition process. Hence, we can consider the node-features of the graphs as constant values in the diffusion process on $G$. For this reason, the structural differences through the diffusion process are only in the the latent adjacency matrix $\mathbf{A}_t$ and latent edge-feature matrix $\mathbf{R}_t$ of the latent graph $G_t$. However, $\mathbf{R}_t$ is deterministically calculated by $\mathbf{A}_t$ for the given initial edge-feature matrix $\mathbf{R}_0$ because $\mathbf{R}_t$ is a matrix consisting of the rows of $\mathbf{R}_0$ selected by $\mathbf{A}_t$. Therefore, if $G_T$ is generated through the complete graph decomposition, the diffusion process on $G$ can be rewritten based on $\mathbf{A}_t \in [0,1]^{|\mathcal{V}_t| \times |\mathcal{V}_t|}$ for the given $\mathbf{R}_0$ as:

$$\log p(G) \geq E_{q(\mathbf{A}_1|\mathbf{A}_0;\mathbf{R}_0)}[\log p(\mathbf{A}_0|\mathbf{A}_1;\mathbf{R}_0)] - D_{\text{KL}}(q(\mathbf{A}_T|\mathbf{A}_0;\mathbf{R}_0)||p(\mathbf{A}_T;\mathbf{R}_0))$$
$$- \sum_{t=2}^{T} E_{q(\mathbf{A}_t|\mathbf{A}_0;\mathbf{R}_0)}\left[D_{\text{KL}}(q(\mathbf{A}_{t-1}|\mathbf{A}_t,\mathbf{A}_0;\mathbf{R}_0)||p(\mathbf{A}_{t-1}|\mathbf{A}_t;\mathbf{R}_0))\right]. \quad (5)$$

DELID assumes $\mathbf{A}_{t,i,j} \sim \mathcal{N}(\mu_{t,i,j}, \sigma_{t,i,j}^2)$ for $t \in \{1, 2, ..., T-1\}$, where $\mathbf{A}_{t,i,j}$ is the $(i,j)$-th element of $\mathbf{A}_t$, and $\mathcal{N}(\mu_{t,i,j}, \sigma_{t,i,j}^2)$ is a normal distribution parameterized by $\mu_{t,i,j}$ and $\sigma_{t,i,j}$. However, since $\mathbf{A}_0$ and $\mathbf{A}_T$, which are the adjacency matrices of $G_0$ and $G_T$ respectively, should be in $\{0,1\}^{|\mathcal{V}| \times |\mathcal{V}|}$, DELID assumes $\mathbf{A}_{t,i,j} \sim \text{Bernoulli}(p_t)$ parameterized by $p_t$ for $t \in \{0, T\}$.

**Decomposing $G$ into $G_T$.** DELID employs a chemically-informed molecular decomposition method based on extended functional groups (EFGs) (Lu et al., 2021) as an implementation of the complete graph decomposition. The EFG-based method has two benefits: 1) It can generate chemically valid substructures. 2) The size of the substructures is automatically adjusted based on chemical knowledge. In chemical science, several other molecular decomposition methods are available for molecular decomposition, such as the junction tree (Jin et al., 2018) and the BRICS decomposition (Liu et al., 2017). However, DELID uses the EFG-based molecular decomposition due to the following two benefits of the EFG-based molecular decomposition: 1) it is an efficient complete graph decomposition that generates chemically-valid subgraphs, and 2) it can capture commonly appeared large molecular substructures beyond the traditional small functional groups. Appendix G empirically shows the benefits of the EFG-based molecular decomposition over other molecular decomposition methods in molecular property prediction.

### 3.2.2 CONDITIONAL DIFFUSION PROCESS ON ELECTRON-LEVEL INFORMATION ($p(\mathbf{s}|G)$)

DELID calculates the unknown electron-aware latent representation $\mathbf{s}$ through a diffusion process guided by the diffusion process on $G$. $\log p(\mathbf{s}|G)$ is given by the variational diffusion model as:

$$\log p(\mathbf{s}|G) = \log p(\mathbf{s}_T|G) + \sum_{t=1}^{T} \log p(\mathbf{s}_{t-1}|\mathbf{s}_t), \quad (6)$$

where $\mathbf{s}_0$ is the latent embedding representing the ground truth electron-level information $\mathbf{s}$, and $\mathbf{s}_T$ is noised electron-level information corresponding to $G_T$. Recall that we define $\mathbf{s}_T$ as pre-calculated electron-level features (properties) of the decomposed substructures $G_T$ of the input molecule $G$ instead of a simple random variable drawn from the standard normal distribution. Detailed descriptions to obtain $\mathbf{s}_T$ will be presented in Section 3.3.

We can derive a computable lower bound of the second term in Eq. (6) by marginalizing it with respect to the latent graph $G_{t-1}$. The lower bound of the second term is given by:

$$\sum_{t=1}^{T} \log p(\mathbf{s}_{t-1}|\mathbf{s}_t, G) \geq \sum_{t=1}^{T} E_{q(G_{t-1}|G_t)} \left[\log p(\mathbf{s}_{t-1}|\mathbf{s}_t, G_{t-1})\right] - \sum_{t=1}^{T} D_{\text{KL}}(q(G_{t-1}|G_t)||p(G_{t-1}|\mathbf{s}_t)). \quad (7)$$

If $G_T$ is generated through the complete graph decomposition, the lower bound of $\log p(\mathbf{s}|G)$ can also be rewritten based on $\mathbf{A}_t$ and $\mathbf{R}_0$ as:

$$\log p(\mathbf{s}|G) \geq \log p(\mathbf{s}_T|\mathbf{A}_0; \mathbf{R}_0) + \sum_{t=1}^{T} E_{q(\mathbf{A}_{t-1}|\mathbf{A}_t; \mathbf{R}_0)}[\log p(\mathbf{s}_{t-1}|\mathbf{s}_t, \mathbf{A}_{t-1}; \mathbf{R}_0)]$$

$$- \sum_{t=1}^{T} D_{\text{KL}}(q(\mathbf{A}_{t-1}|\mathbf{A}_t; \mathbf{R}_0)||p(\mathbf{A}_{t-1}|\mathbf{s}_t; \mathbf{R}_0)). \quad (8)$$

The full derivation of the conditional diffusion process on $\mathbf{s}$ is provided in Appendix B. In the conditional diffusion process, DELID assumes that $p(\mathbf{s}_{t-1}|\mathbf{s}_t, \mathbf{A}_{t-1}; \mathbf{R}_0)$ follows the parameterized normal distributions for $t \in \{2, 3, ..., T\}$, while $p(\mathbf{s}_0|\mathbf{s}_1, \mathbf{A}_0; \mathbf{R}_0)$ and $p(\mathbf{s}_T|\mathbf{A}_0; \mathbf{R}_0)$ are assumed to follow the parameterized Bernoulli distributions.

It is important to note that the conditional representation of the diffusion process on $\mathbf{s}$ described above shows that *we can calculate the lower bound of* $\log p(\mathbf{s}|G)$ *without the ground truth values of* $\mathbf{s}$. Furthermore, the conditional representation also demonstrates that we can maximize $\log p(\mathbf{s}|G)$ by minimizing the KL divergence between $q(\mathbf{A}_{t-1}|\mathbf{A}_t; \mathbf{R}_0)$ and $p(\mathbf{A}_{t-1}|\mathbf{s}_t; \mathbf{R}_0)$ so that the diffusion process conditioned by $\mathbf{s}_t$ follows the diffusion process on $G$.

## 3.3 A Retrieval Process for obtaining $\mathbf{s}_T$ in Conditional Diffusion Process

In Section 3.2.2, we formulated the conditional diffusion process, which can be performed without the ground truth $\mathbf{s}$. However, we still need the ground truth $\mathbf{s}_T$ to perform the conditional diffusion process, since DELID defined $\mathbf{s}_T$ in Eq. (8) as the pre-calculated electron-level features of the decomposed substructures $G_T$ instead of the simple random noise. In this section, we will present a retrieval process of DELID to obtain $\mathbf{s}_T$ without expensive quantum mechanical calculations.

Formally, $\mathbf{s}_T$ is defined as an embedding vector calculated by a trainable neural network for an input matrix $\mathbf{Q} \in \mathbb{R}^{K \times m}$ containing electron-level features about the $K$ decomposed substructures, where the $k$-th row of $\mathbf{Q}$, denoted by $\mathbf{Q}_k$, is the pre-calculated $m$-dimensional electron-level features of the $k$-th substructure $\mathcal{F}_k \in \mathcal{V}_T$. A straightforward way to obtain $\mathbf{Q}$ for calculating $\mathbf{s}_T$ is to execute the quantum mechanical calculations for each $\mathcal{F}_k$. However, researchers in physical science have constructed public chemical databases that provide the electron-level features of small molecules via high-throughput quantum mechanical calculations (Ramakrishnan et al., 2014; Hoja et al., 2021; Kim et al., 2019), and the public chemical databases already provide various electron-level features for most possible small molecules. Hence, DELID leverages the readily accessible databases by taking the pre-calculated electron-level features of the decomposed substructures $\mathcal{F}_1, \mathcal{F}_2, ..., \mathcal{F}_K$ based on a graph matching method for the public chemical databases.

More precisely, $\mathbf{Q}_k$ is determined through the following retrieval process for a given public chemical database $\mathcal{D}_{qm}$, such as QM9 (Ramakrishnan et al., 2014) and PubChemQC (Nakata & Shimazaki, 2017) datasets. The retrieval process is given by:

$$\mathbf{Q}_k = \mathbf{s}_{qm,i^*}, \quad (9)$$

where $\mathbf{s}_{qm,i^*}$ is the pre-calculated electron-level features (e.g., electronic energies) of the $i^*$-th small molecule $G_{qm,i^*}$ in $\mathcal{D}_{qm}$, and an index $i^*$ is calculated by:

$$i^* = \underset{i \in \{1, 2, ..., |\mathcal{D}_{qm}|\}}{\arg\max} \phi(\mathcal{F}_k, G_{qm,i}), \quad (10)$$

$\phi : \mathcal{G} \times \mathcal{G} \rightarrow \mathbb{R}$ is the Tanimoto similarity metric (Bajusz et al., 2015) to calculate the similarity between two molecular graphs $\mathcal{F}_k$ and $G_{qm,i}$ in a graph domain $\mathcal{G}$. In other words, for a decomposed substructure $\mathcal{F}_k$, we assign the pre-calculated electron-level features of a molecule in $\mathcal{D}_{qm}$

whose Tanimoto similarity with $\mathcal{F}_k$ is the highest. By performing the above retrieval process for all $\mathcal{F}_k \in \{\mathcal{F}_1, \mathcal{F}_2, ..., \mathcal{F}_K\}$, we can get $\mathbf{Q}$ for constructing $\mathbf{s}_T$ without expensive quantum mechanical calculations. In the implementation of DELID, we used the QM9 dataset as $\mathcal{D}_{qm}$ and transferred 15 energy- and polarity-related electron-level features for generating $\mathbf{Q}_k$.

## 4 EXPERIMENTS

We compared the prediction capabilities of DELID with those of state-of-the-art methods on various benchmark molecular datasets containing experimentally collected molecules and their properties. We focused on evaluating the prediction capabilities of DELID on the *experimentally generated* datasets rather than *simulated* datasets (e.g., the QM9 dataset) due to the following reasons: 1) The simulated datasets are not suitable to evaluate the prediction capabilities of machine learning methods in real-world chemical applications because most molecules in the simulated datasets are too simple and small (Ramakrishnan et al., 2014; Hoja et al., 2021). 2) Unlike the simulated datasets, heterogeneous and out-of-distribution molecules are common in real-world chemical applications, and we will discuss about this difference in Section 4.1 3) Experimental datasets containing the measurement noises from the uncertainty of the electronic distributions are closer to real-world nature than the simulated datasets (Wu et al., 2018; Joung et al., 2020).

Although we focused on evaluating the prediction capabilities of DELID on the experimental datasets, DELID also showed the prediction accuracy comparable to state-of-the-art methods on a large simulated dataset (Appendix H). In addition to the evaluation on the large simulated dataset, we conducted additional experiments to evaluate the prediction performances of DELID for classification tasks (Appendix I) and for different molecular scales of $\mathcal{D}_{qm}$ (Appendix J).

**Datasets.** We employed nine benchmark molecular datasets constructed by real-world chemical experiments. The benchmark molecular datasets were selected from well-known databases in molecular science (Wu et al., 2018; Wu & Wei, 2018; Mendez et al., 2019; Joung et al., 2020). For comprehensive evaluations, we selected the benchmark molecular datasets from four different chemical applications: physicochemistry, toxicity, pharmacokinetics, and optics. The characteristics of the benchmark molecular datasets are summarized in Appendix D.

**Competitors.** We categorized competitor methods into three classes according to commonly used molecular descriptors: molecular fingerprint, 2D molecular graph, and 3D molecular graph. 1) For the molecular fingerprints, we generated three XGBoost (Chen & Guestrin, 2016) based ensemble methods called XGB-Mor, XGB-FC, and XGB-MK that predict target molecular properties from input Morgan (Mor) (Rogers & Hahn, 2010), functional-class (FC) (Rogers & Hahn, 2010), and MACCS Key (MK) (Singh et al., 2009) fingerprints, respectively. Even though the XGB-based ensemble methods with the molecular fingerprints are simple and trivial, they have shown state-of-the-art prediction accuracy in various chemical applications (Ding et al., 2021; Li et al., 2023b). 2) For the 2D molecular graph, we employed five GNNs: GIN (Xu et al., 2018), EGCN (Tailor et al., 2021), MPNN (Gilmer et al., 2017), D-MPNN (Yang et al., 2019), UniMP (Shi et al., 2021), and AttFP (Xiong et al., 2019). 3) Although 3D-GNNs are not applicable to the experimental molecular datasets, we calculated the 3D atomic coordinates based on FFSEC and evaluated five 3D-GNNs for the input FFSEC-generated 3D graphs. The following 3D GNNs were employed as competitor methods: SchNet (Schütt et al., 2017), DimeNet++ (Gasteiger et al., 2020), PhysChem (Yang et al., 2021), M3GNet (Chen & Ong, 2022), FAENet (Duval et al., 2023), and ConAN (Nguyen et al., 2024). However, we were not able to execute or evaluate several 3D-GNNs (Schütt et al., 2021; Gasteiger et al., 2021; Liao & Smidt, 2022) due to out-of-memory problems or additional requirements on input data. Appendix E provides brief descriptions of the competitor methods.

**Implementations.** We used MPNN and GIN for the GNN-based embedding network of $G$ and $G_T$ in DELID, respectively. The hyperparameters of DELID and competitor methods were optimized by a grid search on commonly used hyperparameter sets. Also, we followed the original implementation of the competitor methods to set method specific hyperparameters. The hyperparameter settings of DELID for each benchmark datasets are given in Appendix F. For the information retrieval of DELID in Section 3.3, we used the QM9 dataset (Ramakrishnan et al., 2014) generated by a high-throughput quantum mechanical calculation on small organic molecules. Instead of the original QM9 dataset, we used a subset of the QM9 dataset containing the molecules with maximum six atoms as $\mathcal{D}_{qm}$ because too large molecules are redundant in matching the decomposed small substructures. The source code of DELID is publicly available at `https://github.com/ngs00/DELID`.

Table 1: The $R^2$-scores of the competitor methods and DELID on benchmark experimental molecular datasets. N/R and N/A mean a negative $R^2$-score indicating a failure of regression and an execution failure related to out of memory or numerical errors, respectively.

| Input Type | Method | Lipop | ESOL | ADMET | IGC50 | LD50 | LC50 | LMC-H | CH-DC | CH-AC |
|---|---|---|---|---|---|---|---|---|---|---|
| Molecular Fingerprint | XGB-Mor | 0.531 (0.024) | 0.659 (0.045) | 0.717 (0.021) | 0.621 (0.040) | 0.390 (0.133) | 0.497 (0.016) | 0.505 (0.018) | N/R | N/R |
| | XGB-FC | 0.578 (0.018) | 0.686 (0.052) | 0.720 (0.009) | 0.628 (0.023) | 0.501 (0.052) | 0.519 (0.025) | 0.503 (0.007) | N/R | N/R |
| | XGB-MK | 0.542 (0.041) | 0.764 (0.047) | 0.761 (0.020) | 0.680 (0.037) | 0.486 (0.112) | 0.526 (0.021) | 0.471 (0.019) | N/R | N/R |
| 3D Molecular Graph | SchNet | 0.667 (0.021) | 0.881 (0.026) | 0.834 (0.012) | 0.765 (0.034) | 0.527 (0.025) | 0.467 (0.024) | 0.456 (0.024) | 0.713 (0.050) | 0.702 (0.037) |
| | DimeNet++ | N/R | 0.878 (0.025) | N/R | 0.779 (0.019) | 0.541 (0.045) | N/A | 0.352 (0.101) | N/A | N/A |
| | PhysChem | 0.694 (0.024) | 0.848 (0.032) | N/A | 0.814 (0.017) | 0.511 (0.053) | N/A | N/A | N/A | N/A |
| | M3GNet | N/A | 0.857 (0.025) | N/A | 0.697 (0.029) | 0.531 (0.034) | N/A | N/A | N/A | N/A |
| | FAENet | 0.670 (0.036) | 0.869 (0.013) | 0.788 (0.020) | 0.708 (0.015) | 0.474 (0.020) | 0.528 (0.094) | 0.437 (0.025) | 0.437 (0.132) | 0.310 (0.136) |
| | ConAN | 0.738 (0.018) | **0.909** (0.015) | **0.845** (0.028) | 0.819 (0.007) | 0.531 (0.041) | 0.572 (0.070) | 0.466 (0.028) | 0.405 (0.108) | 0.388 (0.115) |
| 2D Molecular Graph | GIN | 0.709 (0.019) | 0.808 (0.017) | 0.807 (0.023) | 0.792 (0.015) | 0.545 (0.016) | 0.525 (0.080) | 0.472 (0.033) | 0.242 (0.010) | N/R |
| | EGCN | 0.716 (0.021) | 0.822 (0.029) | 0.814 (0.021) | 0.777 (0.020) | 0.550 (0.018) | 0.503 (0.080) | 0.497 (0.038) | 0.226 (0.086) | N/R |
| | MPNN | 0.727 (0.018) | 0.810 (0.042) | 0.801 (0.028) | 0.764 (0.027) | 0.502 (0.022) | 0.487 (0.108) | 0.461 (0.032) | 0.385 (0.023) | N/R |
| | D-MPNN | 0.726 (0.037) | 0.879 (0.013) | 0.820 (0.018) | 0.787 (0.008) | 0.521 (0.011) | 0.566 (0.098) | 0.494 (0.011) | N/R | N/R |
| | UniMP | 0.718 (0.010) | 0.810 (0.036) | 0.817 (0.018) | 0.756 (0.040) | 0.512 (0.026) | 0.531 (0.078) | 0.478 (0.026) | 0.166 (0.051) | N/R |
| | AttFP | 0.710 (0.021) | **0.909** (0.018) | **0.851** (0.027) | 0.807 (0.013) | 0.513 (0.016) | **0.642** (0.079) | 0.456 (0.031) | 0.441 (0.099) | 0.296 (0.370) |
| | DELID | **0.782** (0.013) | **0.912** (0.014) | 0.834 (0.042) | **0.844** (0.006) | **0.566** (0.024) | **0.644** (0.068) | **0.532** (0.048) | **0.886** (0.035) | **0.885** (0.023) |

## 4.1 MOLECULAR PROPERTY PREDICTION ON REAL-WORLD COMPLEX MOLECULES

We measured the $R^2$-scores of DELID and the competitor methods on the nine benchmark molecular datasets. In this experiment, we focused on evaluating the representation capabilities of DELID and the competitor methods on the experimental datasets containing experimentally collected complex and large molecules rather than the simulated datasets. Note that the $R^2$-score is a normalized metric to measure the regression accuracy. For all datasets, the $R^2$-scores were measured by the 5-fold cross-validation. Table 1 shows the measured $R^2$-scores on the benchmark molecular datasets. Although the 3D-GNNs were sophisticatedly designed to capture the inter-atomic interactions in the 3D geometry and showed accuracy improvements on simulated molecular datasets, they were not executable on many benchmark molecular datasets containing complex and large real-world molecules. However, DELID showed reliable execution performances and achieved state-of-the-art prediction accuracy on most benchmark datasets.

One of the main limitations of the 3D-GNNs is that their generalization capabilities are limited on large molecular graphs due to the impractical time complexities and the easily overfitted embedding schemes (Li et al., 2023a). In the experiments, the $R^2$-scores of the 3D-GNNs were also lower than those of the 2D-GNNs on the Lipop and LMC-H datasets containing many large molecules, even though the 3D-GNNs employ sophisticatedly designed embedding methods with more model parameters. This experimental results directly show the limitations of the 3D-GNNs on real-world complex and large molecules. However, DELID achieved the highest $R^2$-scores on the Lipop and LMC-H datasets. These results show the practical potential of DELID in real-world chemical applications where complex and large molecules commonly appear.

We investigated the reasons for the failure of the competitor methods on the CH-DC and CH-AC datasets. Fig. 3 shows the data distributions of the CH-DC and CH-AC datasets. We also presented the data distributions of the ESOL and IGC50 datasets where most competitor methods successfully learned the relationships between the molecules and the target properties. Additionally, we plotted the data distribution of the QM9 dataset together with the data distributions of the benchmark datasets for comparative analysis on the simulated and experimental molecular datasets. The molecules in the datasets were projected to the 2D space through randomly initialized MPNN to preserve the density of the data.

We observed two major results in Fig. 3. First, the data distribution of the simulated dataset was biased, whereas the data distribution of the experimental datasets covered larger chemical spaces

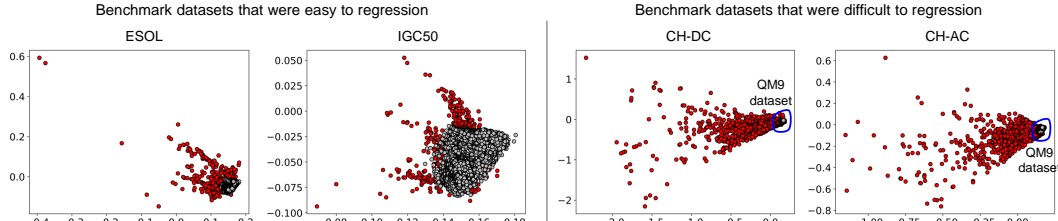

Figure 3: Data Distributions of the benchmark molecular datasets. Black: molecular emebeddings of a simulated dataset called the QM9 dataset. Red: molecular embeddings of an experimentally generated dataset. Note that the data distribution of the QM9 dataset can be plotted differently depending on the scale of the data distribution of the ESOL, IGC50, CH-DC, and CH-AC datasets.

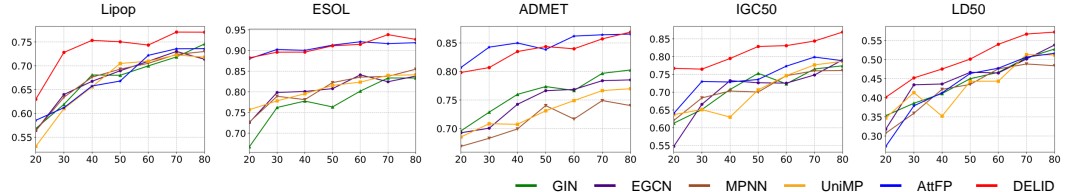

Figure 4: The $R^2$-scores for different sizes of the training data on the Lipop, ESOL, ADMET, IGC50, and LD50 datasets. X-axis: Ratio of the training data. Y-axis: Measured $R^2$-score.

with many outlier molecules. This observation justifies why evaluating machine learning methods on experimentally collected molecular datasets is crucial in validating their practical availability in real-world chemical applications. Second, the CH-DC and CH-AC datasets where most competitor methods failed to predict the target properties covered extremely larger chemical spaces compared to the QM9, ESOL, and IGC50 datasets. Nevertheless, DELID successfully captured the underlying relationships between the molecules and their optical properties on the CH-DC and CH-AC datasets.

## 4.2 PREDICTION ACCURACY ON VARIOUS SIZES OF TRAINING DATA

Since conducting chemical experiments to obtain the experimentally labeled data is time-consuming and labor-intensive, the lack of training data remains one of the main challenges of machine learning in chemical applications (Dou et al., 2023a). As described in Section 3.3, DELID is flexible in incorporating external electron-level features into molecular representation learning, which is beneficial in constructing an accurate prediction model on small training datasets. In this experiment, we compared the $R^2$-scores of DELID and the competitor methods over different sizes of training datasets to demonstrate the effectiveness of DELID on small training datasets.

Fig. 4 shows the $R^2$-scores of DELID and the competitor methods for different sizes of training datasets. We measured the $R^2$-scores on the Lipop, ESOL, ADMET, IGC50, and LD50 datasets in which DELID and most competitor methods achieved the $R^2$-scores greater than 0.5. However, We did not compare the $R^2$-scores of the XGB- and 3D structure-based methods because most of them failed on small training datasets. Obviously, we were able to observe that the prediction accuracy tends to be improved as the size of the training dataset increases for most methods. Among the competitor methods, AttFP showed comparable generalization capabilities to those of DELID on the ESOL and ADMET datasets. However, the accuracy improvements by AttFP were marginal compared to GIN, EGCN, MPNN, and UniMP on the Lipop, IGC50, and LD50 datasets. By contrast, DELID consistently showed better generalization performances regardless of the benchmark datasets compared to GIN, EGCN, MPNN, and UniMP. Furthermore, DELID outperformed AttFP on the Lipop, IGC50, and LD50 datasets. These experimental results show the practical potential of DELID in real-world chemical applications, which usually suffer from the lack of training data.

## 4.3 PREDICTION ACCURACY FOR DIFFERENT ELECTRON-LEVEL FEATURES

The choice of the electron-level features in the information retrieval can affect the representation capabilities of DELID. To evaluate the prediction capability of DELID for different electron-level features, we measured the $R^2$-scores of DELID for different electron-level features provided in the

Table 2: The $R^2$-scores of DELID for different electron-level features.

| Category of Features | Lipop | ESOL | ADMET | IGC50 | LD50 | LC50 | LMC-H | CH-DC | CH-AC |
|---|---|---|---|---|---|---|---|---|---|
| Energy-related features | 0.776 (0.015) | 0.911 (0.013) | 0.835 (0.036) | 0.842 (0.013) | N/R | 0.642 (0.080) | 0.538 (0.029) | 0.784 (0.149) | 0.832 (0.097) |
| Polarity-related features | 0.778 (0.006) | 0.908 (0.011) | 0.755 (0.196) | 0.840 (0.016) | 0.569 (0.020) | 0.632 (0.091) | 0.525 (0.035) | 0.886 (0.008) | 0.877 (0.013) |
| Other features | 0.785 (0.012) | 0.910 (0.012) | 0.711 (0.217) | 0.849 (0.012) | 0.556 (0.018) | 0.632 (0.070) | 0.534 (0.024) | 0.773 (0.147) | 0.861 (0.010) |
| All features (DELID) | 0.782 (0.013) | 0.912 (0.014) | 0.834 (0.042) | 0.844 (0.006) | 0.566 (0.024) | 0.644 (0.068) | 0.532 (0.048) | 0.886 (0.035) | 0.885 (0.023) |

Table 3: The $R^2$-scores of DELID and its three variants in the Ablation study.

| Method | Atom-Level Information | Electron-Level Information | Information Diffusion | Lipop | ESOL | ADMET | IGC50 | LD50 | LC50 | LMC-H | CH-DC | CH-AC |
|---|---|---|---|---|---|---|---|---|---|---|---|---|
| $\text{DELID}_{at}$ | ✓ | ✗ | ✗ | 0.727 (0.018) | 0.810 (0.042) | 0.801 (0.028) | 0.764 (0.027) | 0.502 (0.022) | 0.487 (0.108) | 0.461 (0.032) | 0.385 (0.023) | N/R |
| $\text{DELID}_{et}$ | ✗ | ✓ | ✗ | 0.220 (0.014) | 0.445 (0.060) | 0.537 (0.022) | 0.419 (0.017) | 0.200 (0.024) | 0.226 (0.043) | 0.243 (0.038) | 0.600 (0.032) | 0.548 (0.032) |
| $\text{DELID}_{qm}$ | ✓ | ✓ | ✗ | 0.775 (0.005) | **0.908** (0.014) | **0.824** (0.005) | 0.828 (0.011) | 0.537 (0.030) | 0.616 (0.085) | 0.502 (0.046) | 0.846 (0.026) | **0.875** (0.013) |
| DELID | ✓ | ✓ | ✓ | **0.782** (0.013) | **0.912** (0.014) | **0.834** (0.042) | **0.844** (0.006) | **0.566** (0.024) | **0.644** (0.068) | **0.532** (0.048) | **0.886** (0.035) | **0.885** (0.023) |

QM9 dataset. We categorized the provided electron-level features into the following three classes: 1) energy-related features, 2) polarity-related features, 3) other features. Note that DELID used all electron-level features in representation learning. Table 2 shows the $R^2$-scores of DELID for different kinds of the electron-level features. DELID showed significant improvements on the ADMET and CH-DC datasets by using the energy- and polarity-related electron-level features respectively, because of the direct relationships between the target molecular properties and these electron-level features (Dong et al., 2018; Joung et al., 2020). However, DELID exploiting all electron-level features showed the prediction accuracy comparable to the best models for all datasets, and this result shows that DELID can select important electron-level features for a given target task.

## 4.4 Ablation Study on DELID

We conducted an ablation study to evaluate the effectiveness of the electron-level information and self-supervised diffusion of DELID. We generated three variants of DELID for the ablation study. 1) $\text{DELID}_{at}$ is a model that learns the molecular representations using only the atom-level molecular descriptor $G$, which is the same as MPNN. 2) $\text{DELID}_{et}$ learns the molecular representations using only the fragmented information defined as $G_T$ and $\mathbf{s}_T$. 3) $\text{DELID}_{qm}$ predicts the target molecular property based on Eq. (3) without the self-supervised diffusion on $\mathbf{s}$, i.e., $\text{DELID}_{qm}$ predicts the target molecular property by $y = f(G) + \psi(G_T, \mathbf{s}_T; G)$. Table 3 shows the $R^2$-scores of DELID and its three variants for the ablation study. The $R^2$-scores were measured by the 5-fold cross-validation. The $R^2$-scores of $\text{DELID}_{et}$ shows that the incomplete electron-level features is not sufficient to predict the molecular properties of the original molecules. The accuracy improvements were remarkable in $\text{DELID}_{qm}$ compared to $\text{DELID}_{at}$ and $\text{DELID}_{et}$, and this result shows that integrating the atom-level and electron-level information is crucial for accuracy molecular property prediction. However, we were able to observe further improvements by DELID for all benchmark molecular datasets. These results demonstrate the effectiveness of the electron-aware molecular representation learning based on the self-supervised diffusion.

## 5 Conclusion

This paper proposed DELID to learn informative molecular representations of real-world complex and large molecules based on the self-supervised diffusion process on the electron-level information. In this paper, we mathematically showed that DELID can learn the electron-aware molecular representations by approximating the diffusion process started from the fragmented electron-level information to the diffusion process started from the decomposed substructures, even though the complete electron-level information about the molecules is not known. By employing the self-supervised diffusion, DELID achieved state-of-the-art prediction accuracy on extensive benchmark datasets containing experimentally collected molecules and their molecular properties. The experimental results showed the practical potentials of DELID in real-world chemical applications. As a future work, an efficient method to construct the calculation databases at an accurate calculation level for the coarse-graining representation learning of DELID needs to be considered to provide more accurate electron-level features to DELID.

ACKNOWLEDGMENTS

This work was supported by Ministry of Trade, Industry and Energy (No. TS241-10R) and National Research Foundation of Korea (NRF) funded by Ministry of Science and ICT (NRF-2022M3J6A1063021, RS-2024-00406985).

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

## A  SAMPLING METHOD FOR PARAMETERIZED BERNOULLI DISTRIBUTION

Unlike the conventional diffusion models, DELID assumes the parameterized Bernoulli distribution on $\mathbf{A}_0$ and $\mathbf{A}_T$ because each element of them should be in the binary domain $\{0, 1\}$. Hence, we need to generate sample from reparameterized Bernoulli distribution. By the existing work on categorical reparameterization (Jang et al., 2016; Maddison et al., 2016), we can sample a random variable following the Bernoulli distribution under the reparameterized distribution as:

$$\mathbf{z} = \sigma(\log \epsilon - \log(1 - \epsilon) + \log \phi(\mathbf{x}) - \log(1 - \phi(\mathbf{x}))), \tag{11}$$

where $\sigma$ is the sigmoid function, $\phi \in (0, 1)$ is a trainable shape parameter of Bernoulli distribution, and $\epsilon \sim U(0, 1)$ is a random number from an uniform distribution $U(0, 1)$.

## B  DERIVATION OF THE CONDITIONAL DIFFUSION PROCESS ON $\mathbf{s}$

In this section, we present a detailed derivation of the conditional diffusion process on $\mathbf{s}$. Since the second term $p(\mathbf{s}|G)$ in Eq. (2) is not directly computable, we derive the lower bound of $\log p(\mathbf{s}|G)$ based on a diffusion process started from $\mathbf{s}_T$ as follows.

$$\log p(\mathbf{s}|G) = \log p(\mathbf{s}_T|G) + \sum_{t=1}^{T} \log p(\mathbf{s}_{t-1}|\mathbf{s}_t) \tag{12}$$

We can marginalize the second term in Eq. (12) for $G_{t-1}$ and calculate its lower bound as follows.

$$\sum_{t=1}^{T} \log p(\mathbf{s}_{t-1}|\mathbf{s}_t) = \sum_{t=1}^{T} \log \left( \int_{G_{t-1}} p(\mathbf{s}_{t-1}|\mathbf{s}_t, G_{t-1}) p(G_{t-1}|\mathbf{s}_t) dG_{t-1} \right)$$

$$= \sum_{t=1}^{T} \log \left\{ \int_{G_{t-1}} \left( \frac{p(\mathbf{s}_{t-1}|\mathbf{s}_t, G_{t-1}) p(G_{t-1}|\mathbf{s}_t)}{q(G_{t-1}|G_t)} \right) q(G_{t-1}|G_t) dG_{t-1} \right\}$$

$$\geq \sum_{t=1}^{T} \int_{G_{t-1}} \log \left( \frac{p(\mathbf{s}_{t-1}|\mathbf{s}_t, G_{t-1}) p(G_{t-1}|\mathbf{s}_t)}{q(G_{t-1}|G_t)} \right) q(G_{t-1}|G_t) dG_{t-1}$$

$$= \sum_{t=1}^{T} E_{q(G_{t-1}|G_t)} \left[ \log p(\mathbf{s}_{t-1}|\mathbf{s}_t, G_{t-1}) \right] - \sum_{t=1}^{T} D_{\mathrm{KL}}(q(G_{t-1}|G_t)||p(G_{t-1}|\mathbf{s}_t)). \tag{13}$$

Finally, the lower bound of $\log p(\mathbf{s}|G)$ is given by:

$$\log p(\mathbf{s}|G) \geq \log p(\mathbf{s}_T|G) + \sum_{t=1}^{T} E_{q(G_{t-1}|G_t)} \left[ \log p(\mathbf{s}_{t-1}|\mathbf{s}_t, G_{t-1}) \right]$$

$$- \sum_{t=1}^{T} D_{\mathrm{KL}}(q(G_{t-1}|G_t)||p(G_{t-1}|\mathbf{s}_t)). \tag{14}$$

If $G_T$ is generated through the complete graph decomposition, we also rewrite the lower bound of $\log p(\mathbf{s}|G)$ as follows.

$$\log p(\mathbf{s}|G) \geq \log p(\mathbf{s}_T|\mathbf{A}_0; \mathbf{R}_0) + \sum_{t=1}^{T} E_{q(\mathbf{A}_{t-1}|\mathbf{A}_t; \mathbf{R}_0)} [\log p(\mathbf{s}_{t-1}|\mathbf{s}_t, \mathbf{A}_{t-1}; \mathbf{R}_0)]$$

$$- \sum_{t=1}^{T} D_{\mathrm{KL}}(q(\mathbf{A}_{t-1}|\mathbf{A}_t; \mathbf{R}_0)||p(\mathbf{A}_{t-1}|\mathbf{s}_t; \mathbf{R}_0)). \tag{15}$$

## C  ALGORITHMIC DESCRIPTION OF DELID

Algorithm 1 shows an algorithmic description of the forward and training processes of DELID.

## D  STATISTICS OF THE BENCHMARK EXPERIMENTAL DATASETS

Table 4 shows the statistics and target molecular properties of the nine benchmark molecular datasets used for the experimental evaluations.

---

**Algorithm 1** Training Process of DELID

---

**Input:** $\mathcal{D}_{train}$: a training dataset;
$\qquad\quad$ $\mathcal{D}_{qm}$: a dataset for the information retrieval;
$\qquad\quad$ $\eta$: an initial learning rate of the optimizer;
**Output:** $\theta^*$: optimized model parameters;
**repeat**
$\quad$ **for** $(G, y) \in \mathcal{D}_{train}$ **do**
$\qquad$ // Decomposition of an atom-level molecular graph.
$\qquad$ $\mathcal{F}_1, \mathcal{F}_2, ..., \mathcal{F}_K \leftarrow \texttt{EFGDecomposition}(G)$
$\qquad$ // Information retrieval to obtain $\mathbf{s}_T$.
$\qquad$ **for** $k = 1; k \leq K$ **do**
$\qquad\quad$ $i^* = \arg\max_{i \in \{1,2,...,|\mathcal{D}_{qm}|\}} \phi(\mathcal{F}_k, G_{qm,i})$
$\qquad\quad$ $\mathbf{Q}_k = \mathbf{s}_{qm,i^*}$
$\qquad$ **end for**
$\qquad$ // Model parameter optimization.
$\qquad$ Calculate $J_\theta = \log p_\theta(y, \mathbf{s}, G)$ by Eqs. (3), (5), (8).
$\qquad$ $\theta \leftarrow \theta + \eta \nabla J_\theta$
$\quad$ **end for**
**until** $\theta$ is converged

---

Table 4: Statistics and target molecular properties of the benchmark molecular datasets that contain the atom-level molecular structures and their experimentally observed target properties.

| Application Category | Dataset | Target Molecular Property | # of Molecules | Average Number of Atoms |
|---|---|---|---|---|
| Physicochemistry | Lipop (Mendez et al., 2019) | Lipophilicity | 4,200 | 48.51 |
| | ESOL (Delaney, 2004) | Aqueous solubility | 1,128 | 25.64 |
| | ADMET (Dong et al., 2018) | Aqueous solubility | 4,801 | 26.76 |
| Toxicity | IGC50 (Wu & Wei, 2018) | Tetrahymenapyriformis toxicity | 1,791 | 19.29 |
| | LC50 (Wu & Wei, 2018) | Fathead minnow toxicity | 822 | 22.32 |
| | LD50 (Wu & Wei, 2018) | Oral rat toxicity | 7,412 | 31.30 |
| Pharmacokinetics | LMC-H (Mendez et al., 2019) | Liver microsomal clearance in human | 5,347 | 54.53 |
| Optics | CH-DC (Joung et al., 2020) | Absorption max in Dichloromethane | 2,429 | 28.95 |
| | CH-AC (Joung et al., 2020) | Absorption max in Acetonitrile | 1,781 | 29.71 |

## E  COMPETITOR METHODS

In the experiments, we compared the prediction capabilities of DELID with a baseline tree method and ten state-of-the-art GNNs, which have been widely used in chemical applications. The competitor methods are briefly described as:

- **XGB-Mor**: XGBoost (XGB) (Chen & Guestrin, 2016) is a tree-based gradient boosting model, and it showed state-of-the-art performances in various scientific applications. For the experimental evaluations, we generated XGB-Mor that predicts the target molecular properties for the Morgan (Mor) fingerprints of the atom-level molecular structures (Rogers & Hahn, 2010).

- **XGB-FC**: We generated XGB-FC by combining XGB with the functional-class (FC) fingerprints of the input molecules (Rogers & Hahn, 2010). The FC fingerprint represents the atom-level molecular structures based on their functional substructures and atoms.

- **XGB-MK**: We also generated XGB-MK based on the MACCS Key (MK) fingerprint (Singh et al., 2009), which is one of the most commonly used molecular representations. MACCS key encodes the atom-level molecular structures based on 166-bits binary patterns.

- **GIN** (Xu et al., 2018): Graph isomorphism network (GIN) is an effective framework for graph representation learning based on graph isomorphism test.

- **EGCN** (Tailor et al., 2021): Efficient graph convolution (EGC) is an isotropic GNN based on adaptive filters and aggregation fusion in the node aggregation phase. EGC outperformed common anisotropic GNNs, such as graph attention networks, on benchmark datasets.

- **MPNN** (Gilmer et al., 2017): Message passing neural network is a unified framework of node and edge convolution methods for learning molecular representations on quantum chemistry.

- **D-MPNN** (Yang et al., 2019): Directed MPNN (D-MPNN) is an extension of the original MPNN for the directed molecular graphs. It employs a message passing scheme via directed edges (bonds).

- **UniMP** (Shi et al., 2021): Unified message passing (UniMP) is a transformer-based GNN. UniMP showed state-of-the-art prediction capabilities by incorporating feature and label propagation at both training and inference time based on the transformer architecture.

- **AttFP** (Xiong et al., 2019): AttFP is a network that uses a graph self-attention mechanism to learn molecular representations for drug discovery. AttFP was designed to learn non-local intra-molecular interactions to extract informative molecular representations.

- **SchNet** (Schütt et al., 2017): It is a convolutional neural network for learning molecular representations based on quantum interactions in molecules. It has been widely used as a baseline model in various chemical applications (Schütt et al., 2017; 2018).

- **DimeNet++** (Gasteiger et al., 2020): DimeNet aims to learn molecular representations based on the directional embedding that extracts inter-atomic 3D geometry. DimNet++ is an advanced version of DimeNet to learn the molecular representatios based on the uncertainty-aware directional embedding.

- **PhysChem** (Yang et al., 2021): PhysChem is a neural architecture that learns molecular representations via fusing the information about the inter-atomic geometry and message passing through chemical bonds. PhysChem showed state-of-the-art prediction accuracy on various simulated molecular datasets.

- **M3GNet** (Chen & Ong, 2022): Graph neural networks with three-body interactions (M3GNet) is a neural network to learn molecular representations based on the three-body inter-atomic interactions. Although M3GNet requires large computational costs to calculate the three-body interactions, it showed state-of-the-art prediction accuracy on various molecular and materials datasets.

- **FAENet** (Duval et al., 2023): Frame averaging equivariant GNN (FAENet) is simple and fast GNN optimized for stochastic frame-averaging. FAENet can learn molecular representations by processing atom-relative positions with full flexibility without symmetry-preserving requirements.

- **ConAN** (Nguyen et al., 2024): Conformer aggregation network (ConAN) is an E(3)-invariant molecular conformer aggregation network to learn molecular representations based on ensembles methods on molecular conformers. It showed state-of-the-art prediction accuracy on several molecular datasets by employing a 2D–3D aggregation mechanism based on a differentiable solver for the Fused Gromov-Wasserstein Barycenter problem.

## F  IMPLEMENTATION DETAILS AND HYPERPARAMETER SETTINGS

We followed a common graph-based descriptor to convert an atom-level molecular structure into an attributed graph $G = (\mathcal{V}, \mathcal{U}, \mathbf{A}, \mathbf{X}, \mathbf{R})$, where $\mathcal{V}$ is a set of nodes (i.e., atoms), $\mathcal{U}$ is a set of edges (i.e., chemical bonds), $\mathbf{A} \in \{0, 1\}^{|\mathcal{V}| \times |\mathcal{V}|}$ is an adjacency matrix, $\mathbf{X} \in \mathbb{R}^{|\mathcal{V}| \times d}$ is a $d$-dimensional node-feature matrix, and $\mathbf{R} \in \mathbb{R}^{|\mathcal{U}| \times r}$ is an $r$-dimensional edge-feature matrix (Wieder et al., 2020). We used the pre-defined 200-dimensional atomic embeddings (Goodall & Lee, 2020) to construct the node-feature matrix $\mathbf{X}$. We defined the edge features as an one-hot encoding of 22 bond types (Wieder et al., 2020; Chen et al., 2019). The pre-defined bond types were provided in RDKit[1], which is a popular cheminformatics library in computational chemistry.

The model parameters of DELID were optimized by the AdamW optimizer (Loshchilov & Hutter, 2017) for all experiments in this paper. The initial learning rate and $L_2$ regularization coefficients were fixed to 5e-4 and 5e-6 for all benchmark datasets, respectively. Batch size is also fixed to 64 for all benchmark datasets. The GNN-based embedding networks were constructed by two node

---

[1]https://www.rdkit.org

Table 5: Electron-level features used for the retrieval process of DELID on the QM9 dataset.

| Category | Feature Name | Unit |
|---|---|---|
| Energy-related feature | HOMO | eV |
| | LUMO | eV |
| | HOMO-LUMO gap | eV |
| | Zero point vibrational energy | eV |
| | Internal energy at 0 K | eV |
| | Internal energy at 298.15 K | eV |
| | Enthalpy at 298.15 K | eV |
| | Free energy at 298.15 K | eV |
| | Heat capacity at 298.15 K | eV |
| Polarity-related feature | Dipole moment | Debye |
| | Isotropic polarizability | $Bohr^3$ |
| | Electronic spatial extent | $Bohr^2$ |
| Other features | Rotational constant A | GHz |
| | Rotational constant B | GHz |
| | Rotational constant C | GHz |

aggregation layers and one dense layer with 64 output channels. DELID and experiment scripts were implemented with PyTorch 2.0.0+cu117[2] and PyTorch Geometric 2.3.1[3] under Python 3.9.

In the implementation of the information retrieval on **s**, we used a subset of the QM9 dataset (Ramakrishnan et al., 2014) containing the molecules of maximum six atoms as an external quantum mechanics dataset $\mathcal{D}_{qm}$. We used 15 electron-level features in the QM9 dataset to construct the feature matrix of the fragmented electron-level information **S**. The selected 15 electron-level features of the QM9 dataset are shown in Table 5.

## G    PREDICTION ACCURACY FOR DIFFERENT DECOMPOSITION METHODS

DELID employs the EFG-based decomposition method for generating the decomposed substructures $G_T$ from the input molecule $G$. Table 6 shows the $R^2$-scores of two variants of DELID that use two well-known molecular decomposition methods: BRICS (Liu et al., 2017) decomposition and junction tree method (Jin et al., 2018). Although the $R^2$-scores of DELID were not significantly changed for the implementations of the molecular decomposition methods, DELID with the EFG-based decomposition showed higher $R^2$-scores for most benchmark experimental datasets.

Table 6: The $R^2$-scores of DELID for different molecular decomposition methods.

| Decomposition Method | Lipop | ESOL | ADMET | IGC50 | LD50 | LC50 | LMC-H | CH-DC | CH-AC |
|---|---|---|---|---|---|---|---|---|---|
| BRICS (Liu et al., 2017) | 0.763 (0.018) | 0.908 (0.008) | 0.811 (0.050) | 0.824 (0.008) | 0.516 (0.031) | 0.635 (0.058) | 0.515 (0.040) | 0.865 (0.047) | 0.863 (0.031) |
| Junction Tree (Jin et al., 2018) | 0.770 (0.012) | 0.905 (0.013) | 0.808 (0.033) | 0.823 (0.005) | 0.518 (0.023) | 0.622 (0.057) | 0.518 (0.038) | 0.866 (0.053) | 0.867 (0.031) |
| EFG (Lu et al., 2021) | 0.782 (0.013) | 0.912 (0.014) | 0.834 (0.042) | 0.844 (0.006) | 0.566 (0.024) | 0.644 (0.068) | 0.532 (0.048) | 0.886 (0.035) | 0.885 (0.023) |

## H    PREDICTION ACCURACY ON LARGE SIMULATED DATASETS

Although the calculated physical and chemical properties of the molecules are not reliable in real-world complex molecules, we conducted an experiment of predicting molecular properties on a large simulated dataset to evaluate the prediction capabilities of DELID on large molecular datasets. For the evaluation, we used the QM-GW dataset (Fediai et al., 2023) containing the GW-level HOMO-LUMO gaps of 133,885 molecules. The GW method (Reining, 2018) is an approximation method for the density functional theory calculations. The GW method is computationally expensive but accurate in calculating the molecular properties related to the electronic energies. Thus, the machine learning methods should be able to learn the GW-level calculations on a huge number of molecules in order to build an accurate prediction model on the QM-GW dataset.

---

[2]https://pytorch.org
[3]https://pytorch-geometric.readthedocs.io

Table 7: The $R^2$-scores of DELID and the competitor 2D GNNs on the QM-GW dataset containing 13k molecules and their calculated properties.

| GIN | EGCN | MPNN | D-MPNN | UniMP | AttFP | DELID |
|---|---|---|---|---|---|---|
| 0.863 | 0.880 | 0.880 | 0.879 | 0.878 | 0.862 | **0.885** |
| (0.008) | (0.004) | (0.005) | (0.003) | (0.006) | (0.002) | **(0.003)** |

Table 7 shows the $R^2$-scores of DELID and the competitor methods on the QM-GW dataset (Fediai et al., 2023) containing the GW-level HOMO-LUMO gaps of 133,885 molecules. In this experiment, DELID and all competitor 2D GNNs easily achieved the $R^2$-scores greater than 0.85 because the simulated datasets are generated by the simple and consistent methods. In particular, although some GNNs failed to predict the molecular properties on the experimental datasets, they also achieved the $R^2$-scores greater than 0.85 on the large simulated dataset. This result demonstrates our main argument that the prediction capabilities of the machine learning models on the simulated datasets do not tell us the actual prediction capabilities of the machine learning models in real-world chemical applications.

## I   PREDICTION ACCURACY IN CLASSIFICATION TASKS

In the experiments, we focused on evaluating the prediction performances of the machine learning methods in regression problems due to the following two reasons: 1) The regression problems is a generalized problem of the classification problem, i.e., the classification problem is a specific case of the regression problem, where the number of possible classes in the target variable is fixed to a countable natural number. 2) Most classification problems in physical and chemical applications are fundamentally the downstream tasks of the regression problem.

Table 8: The F1-scores of DELID and the competitor 2D GNNs in the classification tasks of the BACE and BBBP datasets.

| Dataset | GIN | EGCN | MPNN | D-MPNN | CGCNN | UniMP | AttFP | DELID |
|---|---|---|---|---|---|---|---|---|
| BACE | 0.777 | 0.765 | 0.771 | 0.769 | 0.770 | 0.773 | 0.773 | **0.805** |
| | (0.019) | (0.026) | (0.027) | (0.027) | (0.024) | (0.017) | (0.013) | **(0.014)** |
| BBBP | 0.908 | 0.911 | 0.897 | 0.894 | 0.894 | 0.912 | 0.905 | **0.924** |
| | (0.012) | (0.009) | (0.020) | (0.014) | (0.009) | (0.010) | (0.008) | **(0.004)** |

There is no implementation issue of DELID in the classification tasks. Eq. (2) is generally applicable to both regression and classification tasks. In this experiment, we measured the F1-scores of DELID and the competitor 2D-GNNs in the classification tasks. We used two experimentally collected molecular datasets called BACE (Yan & Vassar, 2014) and BBBP (Wu et al., 2018). The BACE and BBBP dataset contain experimentally measured biological activities of 1,513 and 2,050 molecules, respectively. Table 8 shows the measured F1-scores on the BACE and BBBP datasets, and DELID still showed the highest prediction accuracy.

## J   PREDICTION ACCURACY FOR DIFFERENT MOLECULAR SCALES OF EXTERNAL CALCULATION DATASETS

We measured the $R^2$-scores of DELID for different $\mathcal{D}_{qm}$ containing different sizes of small molecules. We generated $\mathcal{D}_{qm,c}$ for $c = \{4, 5, 6, 7\}$, where $c$ is the maximum number of atoms. For example, $\mathcal{D}_{qm,c}$ is constructed from the QM9 dataset by collecting small molecules containing the atoms less than or equal to $c$. Table 9 presents the measured $R^2$-scores of DELID for different values of $c$. DELID showed consistent $R^2$-scores for different sizes of the molecules in $\mathcal{D}_{qm}$ because the input complex and large molecules are already decomposed into the small substructures by the EFG-based decompsotion. This result shows that DELID is robust to the volume of $\mathcal{D}_{qm}$ required for the information retrieval.

Table 9: The $R^2$-scores of DELID on the benchmark molecular datasets for different $\mathcal{D}_{qm}$ containing different sizes of small molecules.

| Max. Num. Atoms $(= c)$ | Num. Molecules in $\mathcal{D}_{qm}$ | Lipop | ESOL | ADMET | IGC50 | LD50 | LC50 | LMC-H | CH-DC | CH-AC |
|---|---|---|---|---|---|---|---|---|---|---|
| $c = 4$ | 45 | 0.780 (0.010) | 0.921 (0.009) | 0.843 (0.029) | 0.863 (0.004) | 0.565 (0.018) | 0.636 (0.093) | 0.533 (0.020) | 0.876 (0.012) | 0.841 (0.066) |
| $c = 5$ | 175 | 0.776 (0.013) | 0.912 (0.010) | 0.680 (0.366) | 0.848 (0.010) | 0.586 (0.022) | 0.641 (0.071) | 0.538 (0.041) | 0.873 (0.015) | 0.880 (0.014) |
| $c = 6$ | 682 | 0.782 (0.013) | 0.912 (0.014) | 0.834 (0.042) | 0.844 (0.006) | 0.566 (0.024) | 0.644 (0.068) | 0.532 (0.048) | 0.886 (0.035) | 0.885 (0.023) |
| $c = 7$ | 3,990 | 0.781 (0.016) | 0.916 (0.012) | 0.843 (0.023) | 0.844 (0.015) | 0.574 (0.020) | 0.658 (0.061) | 0.530 (0.033) | 0.872 (0.022) | 0.882 (0.012) |

## K EXECUTION TIME OF THE TRAINING AND INFERENCE PROCESSES OF DELID

Compared to the conventional GNNs, DELID requires additional computation to execute the information retrieval and the self-supervised diffusion. In this experiment, we compared the execution time of DELID with those of the competitor GNNs. We separated the entire execution process of machine learning methods into data pre-processing, training, and inference steps. We measured the entire execution time of the data pre-processing and inference processes, whereas we measured the execution time of one epoch for the training process. We compared the execution time of GIN, MPNN, UniMP, SchNet, PhysChem, and DELID, as shown in Table 10. The execution time was measured in a machine with Intel i9-12900K CPU, 128G memory, and NVIDIA GeForce RTX 3090 Ti GPU.

Table 10: Execution time of DELID and the competitor methods on the Lipop dataset. The execution time was measured in seconds.

| Category | Method | Data Pre-processing | Training | Inference | Total |
|---|---|---|---|---|---|
| 3D-GNN | SchNet | 119.937 | 1,109.350 | 0.172 | 1,229.459 |
| | PhysChem | 59.453 | 109,271.543 | 10.289 | 109,341.285 |
| 2D-GNN | GIN | 3.109 | 152.504 | 0.031 | 155.644 |
| | MPNN | 3.109 | 429.524 | 0.062 | 432.695 |
| | AttFP | 3.109 | 125.273 | 0.031 | 128.413 |
| | DELID | 50.875 | 562.549 | 0.124 | 613.548 |

The 2D-GNNs were the most efficient among the competitor methods and DELID because they do not use additional molecular descriptors and complex node aggregation mechanisms. In contrast, the 3D-GNNs required the most execution time. In particular, the total execution time of PhysmChem was 109,341 seconds, which is 178 times greater than the total execution time of DELID. Although DELID requires more execution time in the data pre-processing and the conditional diffusion process, it showed comparable execution time with vanilla MPNN, which was employed to implement the atom-level embedding network of DELID.

## L EMBEDDING RESULTS AND MOLECULAR WEIGHT

We visualized the electron-aware representation **s** for the molecular weight, which can be one of the underlying variable determining several target molecular properties. We plotted **s** for the molecular weight on the Lipop, ESOL, and ADMET datasets, where the target molecular properties are related to the molecular weight. As shown in Fig. 5, even though the molecular weight was not provided for training DELID, DELID generated **s** that roughly describes the underlying molecular weight.

## M MEAN ABSOLUTE ERRORS IN MOLECULAR PROPERTY PREDICTION

Table 11 shows maen absolute error (MAE) of the competitor methods and DELID in predicting molecular properties on the experimental molecular datasets. The evaluation results of DELID were consistent with the evaluation reulsts in Talbe 1.

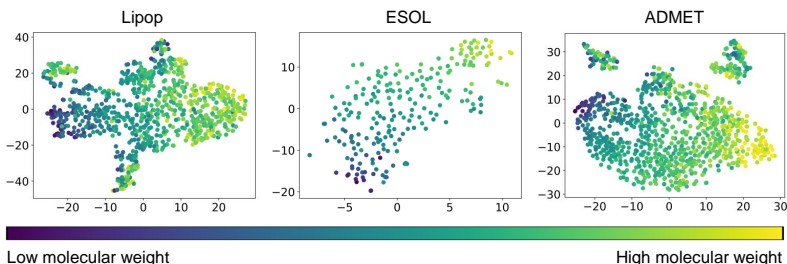

Figure 5: Visualization of the electron-aware representation **s** for the underlying molecular weight.

Table 11: MAEs of the competitor methods and DELID in predicting molecular properties.

| Method | Lipop | ESOL | ADMET | IGC50 | LD50 | LC50 | LMC-H | CH-DC | CH-AC |
|--------|-------|------|-------|-------|------|------|-------|-------|-------|
| GIN | 0.456 (0.010) | 0.597 (0.042) | 0.628 (0.026) | 0.330 (0.020) | 0.443 (0.007) | 0.717 (0.029) | 0.349 (0.006) | 37.268 (2.818) | 33.125 (2.835) |
| EGCN | 0.447 (0.014) | 0.594 (0.045) | 0.602 (0.022) | 0.331 (0.021) | 0.446 (0.009) | 0.722 (0.050) | 0.337 (0.008) | 33.469 (1.462) | 34.578 (2.876) |
| MPNN | 0.434 (0.012) | 0.603 (0.048) | 0.649 (0.022) | 0.351 (0.004) | 0.460 (0.007) | 0.704 (0.073) | 0.342 (0.007) | 31.257 (1.570) | 32.765 (2.870) |
| D-MPNN | 0.422 (0.011) | 0.507 (0.031) | 0.636 (0.019) | 0.327 (0.018) | 0.472 (0.010) | 0.645 (0.056) | 3.332 (0.009) | 40.549 (2.524) | 33.574 (3.042) |
| UniMP | 0.453 (0.020) | 0.609 (0.044) | 0.619 (0.019) | 0.335 (0.020) | 0.450 (0.005) | 0.707 (0.051) | 0.352 (0.010) | 38.546 (2.547) | 35.896 (3.204) |
| AttFP | 0.466 (0.006) | **0.441** (**0.034**) | **0.577** (**0.024**) | 0.315 (0.011) | 0.472 (0.004) | 0.635 (0.062) | 0.335 (0.006) | 54.234 (3.334) | 46.211 (3.303) |
| DELID | **0.395** (**0.011**) | **0.425** (**0.027**) | **0.562** (**0.028**) | **0.279** (**0.004**) | **0.443** (**0.007**) | **0.592** (**0.037**) | **0.310** (**0.007**) | **20.430** (**2.203**) | **19.503** (**1.860**) |

## N   DIFFUSION PROCESS ON **S** IN UNSUPERVISED SETTINGS

In our problem setting in Eq. (2), the entire model parameters of DELID is optimized to maximize $\log p(y, \mathbf{s}, G)$. For this reason, although the diffusion model on **s** is trained by the self-supervised scheme guided by the diffusion model on $G$, the embedding results on **s** are finally affected by the target molecular property $y$. To investigate the representation learning capabilities of DELID in unsupervised settings, we re-implemented DELID to maximize $\log p(\mathbf{s}, G)$ by removing the prediction layers in DELID and measured the Wasserstein distance between the data distributions of the original molecular graph $G$ and the diffusion output $\mathbf{s}_0$ on the ESOL dataset. $G$ was projected into the vector space through randomly initialized GNNs to preserve the data distribution of the original molecular graphs Schneider & Vlachos (2017); Bingham & Mannila (2001).

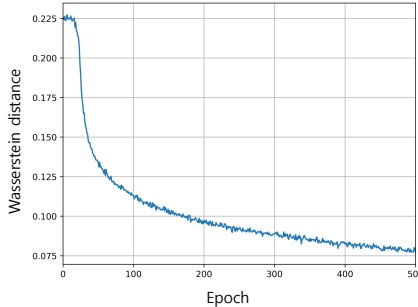

Figure 6: Wasserstein distance between the data distributions of $G$ and $\mathbf{s}_0$ on the ESOL dataset.

As shown Fig. 6, the Wasserstein distance between $G$ and $\mathbf{s}_0$ consistently decreased as the diffusion model on **s** was optimized. This result shows that the diffusion models of DELID worked well in the unsupervised setting. Furthermore, this result is worth noting because the diffusion model on **s** successfully learned the data distribution of $G$ with only the fragmented information about decomposed substructures and their electron-level features.

Table 12: The $R^2$-scores of the competitor methods, DELID, and NN-DELID on benchmark experimental molecular datasets.

| Input Type | Method | Lipop | ESOL | ADMET | IGC50 | LD50 | LC50 | LMC-H | CH-DC | CH-AC |
|---|---|---|---|---|---|---|---|---|---|---|
| Molecular Fingerprint | XGB-Mor | 0.531 (0.024) | 0.659 (0.045) | 0.717 (0.021) | 0.621 (0.040) | 0.390 (0.133) | 0.497 (0.016) | 0.505 (0.018) | N/R | N/R |
| | XGB-FC | 0.578 (0.018) | 0.686 (0.052) | 0.720 (0.009) | 0.628 (0.023) | 0.501 (0.052) | 0.519 (0.007) | 0.503 (0.007) | N/R | N/R |
| | XGB-MK | 0.542 (0.041) | 0.764 (0.047) | 0.761 (0.020) | 0.680 (0.037) | 0.486 (0.112) | 0.526 (0.021) | 0.471 (0.019) | N/R | N/R |
| 3D Molecular Graph | SchNet | 0.667 (0.021) | 0.881 (0.026) | 0.834 (0.012) | 0.765 (0.034) | 0.527 (0.062) | 0.467 (0.025) | 0.456 (0.024) | 0.713 (0.050) | 0.702 (0.037) |
| | DimeNet++ | N/R | 0.878 (0.025) | N/R | 0.779 (0.019) | 0.541 (0.045) | N/A | 0.352 (0.101) | N/A | N/A |
| | PhysChem | 0.694 (0.024) | 0.848 (0.032) | N/A | 0.814 (0.017) | 0.511 (0.053) | N/A | N/A | N/A | N/A |
| | M3GNet | N/A | 0.857 (0.025) | N/A | 0.697 (0.029) | 0.531 (0.034) | N/A | N/A | N/A | N/A |
| | FAENet | 0.670 (0.036) | 0.869 (0.013) | 0.788 (0.020) | 0.708 (0.015) | 0.474 (0.094) | 0.528 (0.025) | 0.437 (0.025) | 0.437 (0.132) | 0.310 (0.136) |
| | ConAN | 0.738 (0.018) | **0.909** (0.015) | **0.845** (0.028) | 0.819 (0.007) | 0.531 (0.041) | 0.572 (0.070) | 0.466 (0.028) | 0.405 (0.108) | 0.388 (0.115) |
| 2D Molecular Graph | GIN | 0.709 (0.019) | 0.808 (0.017) | 0.807 (0.023) | 0.792 (0.015) | 0.545 (0.016) | 0.525 (0.080) | 0.472 (0.033) | 0.242 (0.010) | N/R |
| | EGCN | 0.716 (0.021) | 0.822 (0.029) | 0.814 (0.021) | 0.777 (0.020) | 0.550 (0.018) | 0.503 (0.080) | 0.497 (0.038) | 0.226 (0.086) | N/R |
| | MPNN | 0.727 (0.018) | 0.810 (0.042) | 0.801 (0.028) | 0.764 (0.027) | 0.502 (0.022) | 0.487 (0.108) | 0.461 (0.032) | 0.385 (0.023) | N/R |
| | D-MPNN | 0.726 (0.037) | 0.879 (0.013) | 0.820 (0.018) | 0.787 (0.008) | 0.521 (0.011) | 0.566 (0.098) | 0.494 (0.011) | N/R | N/R |
| | UniMP | 0.718 (0.010) | 0.810 (0.036) | 0.817 (0.018) | 0.756 (0.040) | 0.512 (0.026) | 0.531 (0.078) | 0.478 (0.026) | 0.166 (0.051) | N/R |
| | AttFP | 0.710 (0.021) | **0.909** (0.018) | **0.851** (0.027) | 0.807 (0.013) | 0.513 (0.016) | **0.642** (0.079) | 0.456 (0.027) | 0.441 (0.099) | 0.296 (0.370) |
| | NN-DELID | **0.773** (0.019) | **0.908** (0.009) | 0.823 (0.027) | **0.837** (0.010) | **0.574** (0.026) | **0.634** (0.072) | 0.517 (0.031) | 0.829 (0.057) | **0.860** (0.027) |
| | DELID | **0.782** (0.013) | **0.912** (0.014) | 0.834 (0.042) | **0.844** (0.006) | 0.566 (0.024) | **0.644** (0.068) | **0.532** (0.048) | **0.886** (0.035) | **0.885** (0.023) |

## O    REPRESENTATION LEARNING CAPABILITIES OF DELID IN UNSUPERVISED SETTINGS

DELID is designed to learn electron-aware molecular representations for given molecular structures $G$ and target molecular properties $y$ by maximizing $\log p(y, \mathbf{s}, G)$, as described in Eq. (2). However, the diffusion models on $G$ and $\mathbf{s}$ can be trained without $y$ by maximizing $\log p(\mathbf{s}, G)$ instead of $\log p(y, \mathbf{s}, G)$. In other words, we can train the diffusion models with a fully unsupervised setting and build a prediction model by transferring the diffusion models into the downstream prediction task. In this experiment, we measured the $R^2$-scores of a fully connected neural network (FCNN) that employs the molecular representation of the unsupervised DELID as its input data in molecular property prediction. We denote the FCNN following DELID by NN-DELID.

Table 12 shows the $R^2$-scores of the competitor methods and NN-DELID. In this experiment, NN-DELID also outperformed the competitor methods for most benchmark datasets. However, the $R^2$-scores of DELID were usually higher than those of NN-DELID.

## P    LIMITATIONS AND FUTURE WORK

Since the self-supervised diffusion processes of DELID start from the decomposed molecular substructures and their electron-level features, the performances of DELID are basically dependent on the molecular decomposition methods and the external calculation datasets. In particular, as shown in Table 2, the input electron-level features can directly affect the prediction accuracy of the final prediction models. However, we did not develop a molecular decomposition method and calculation dataset specialized in the self-diffusion processes of DELID. For this reason, the improvements by DLIED are essentially limited to the performances of the EFG-based decomposition method and the QM9 dataset. Therefore, a molecular decomposition and calculation dataset specialized in DELID need to be considered as future work.

