# OpenReview forum: "Self-Supervised Diffusion Models for Electron-Aware Molecular Representation Learning"
_ICLR.cc/2025/Conference — ICLR 2025 Poster_

### Official Review · Reviewer_bQLK · 2024-10-29

**Soundness:** 4
**Presentation:** 4
**Contribution:** 3
**Rating:** 6
**Confidence:** 4

**Summary:**

This paper proposes a method that uses two diffusion models to learn the representations of molecules and electron information, then the representations are used for property predictions. The novelty of this work is to estimate electron information from existing public database, without expensive ground truth labeling, which is achieved by a self supervised training from the molecular diffusion model.

The authors performed comprehensive experiments with relevant baseline models, and demonstrate the state of the art performance of the proposed model on the experimental dataset that contains large molecules and experimental measurements.

**Strengths:**

- Originality: This paper proposes a novel approach that utilizes the accessible electron information of subgraphs (or building blocks) of a molecule from public database, to enhance the representation and property prediction. The self-supervised learning also shows a good theoretical and algorithmic ground, so as to solve the difficulties of the need for expensive calculations for ground truth electronic structure.
- Quality: This paper has demonstrated solid theoretical derivations, and has performed comprehensive numerical experiments on multiple tasks, comparisons with other models and ablation studies.
- Clarity: The figures and formulations are clear and illustrative. The text has good clarity.
- Significance: This work shows a potential way for addressing small scale molecular dataset.

**Weaknesses:**

- Even though it is a good use of publicly available electronic structure for subgraphs/fragments, the final electronic information $s_0$ is only an estimation of the ground truth. And in fact, it is not guaranteed to be a good estimation, as the electronic structure of a molecule with different fragments covalently bonded can be very different from mixing the electronic structures of different individual isolated fragments. Therefore, physically, this self supervised learning, which is essentially trying to mix the electronic information of fragments in the same way as mixing the structural information, is not very meaningful.
- Based on the point above, I would expect that the information mostly gained from the electronic representation will come directly from $s_T$ which are the electronic features of the fragments, instead of from $s_0$. And this can be verified from Table 3, the performance difference between "DELID_qm" and "DELID" is actually very small and generally within the error range.
- The benchmark dataset used here is a small dataset, as it is an experimental dataset and also includes large molecules. But for some other competitor models, it makes sense that they don't perform that well if they are trained from scratch for this small dataset. That does not mean that the method proposed here is really SOTA, as I would expect those competitor method can potentially perform very well with a pretrain-finetune workflow (i.e. pretrained on a large scale simulated datasets, and then finetuned for the small experimental dataset).

**Questions:**

- It is not so clear to me what parts are built with a separate neural network model. For example, my understanding is $\log p(A_{t-1}|A_t)$, $\log p(s_{t-1}|s_t, A_{t-1})$, $\log p(A_{t-1}|s_t)$ are parametrized as separate models. It will be good if the authors can clarify and improve the notations.

---

> ### Author Response · Authors · 2024-11-20
>
> We sincerely appreciate the reviewer’s constructive feedback on our paper!
>
> **W1:** We understand the reviewer’s concerns about the physical meaning of s_0. However, as mentioned in Section 3.1, the purpose of learning s is to learn the uncertainty derived from the electronic structures or densities. Physically, atomic systems essentially contain the electron-derived uncertainty [1]. However, existing machine learning methods overlooked this fact and tried to learn deterministic molecular representations. As described in Eq. (3), DELID was devised to predict molecular properties by considering the electron-derived uncertainty. To this end, we devised the conditional diffusion model to estimate s without expensive quantum mechanical calculations on real-world complex molecules.
>
> DELID employs GNN to parameterize the diffusion probability on G_T containing the decomposed substructures as the nodes, as described in lines 227-230. The GNN aggregates the structural information and assigned electron-level features of the substructures to generate the embedding vector describing the entire molecule. In other words, DELID learns a generalized scheme for aggregating the decomposed substructures and their electron-level information. Appendix M in the uploaded rebuttal file shows that DELID generated the electron-aware representation s highly related to the target molecular properties, even though s is essentially generated without the complete electron-level information about the input molecules.
>
> In addition to the physical meaning, we would like to emphasize the novelty of DELID from a chemical perspective. Chemically, substructures can dominate the entire molecular properties, such as toxicity, solubility, and optical properties. DELID provides a framework to leverage and integrate the atom- and electron-level descriptors of the substructures without additional quantum mechanical calculations. The significant accuracy improvements on the Lipop, IGC50, LD50, LMC-H, CH-DC, and CH-AC datasets empirically show the effectiveness of DELID.
>
> [1] Uncertainty quantification in chemical systems. 2009. Int. J. Numer. Methods Eng.
>
> ---
> **W2:** As mentioned by the reviewer, most of the information is gained from s_T because it is the input feature for calculating s_0. DELID employs multiple GNNs connected by the diffusion process to learn a generalized scheme for aggregating the decomposed substructures (fragments) of the input molecules. As answered the first comment, DELID employs GNN to learn the electron-aware molecular representation from the fragments by capturing both their information and interactions.
>
> Although the accuracy improvements by the conditional diffusion process were margin on some datasets, we would like to remark that DELID showed consistent accuracy improvements regardless of the datasets. Besides, DELID showed accuracy improvements over the error range on the IGC50, LD50, and CH-DC datasets. Please consider not only the prediction accuracy, but also the practical potential of DELID, which was shown in the consistent accuracy improvements regardless of the datasets.
>
> ---
> **W3:** We have two answers for the reviewer’s concern as follows.
> - As shown in the CH-DC and CH-AC datasets in Fig. 3, the calculation dataset (QM9) can cover only small ranges of real-world complex molecules due to the extensive computational costs of the quantum mechanical calculations. Therefore, we believe that a pre-training strategy on the calculation datasets may not be a fundamental solution for building accurate prediction models on experimental molecular datasets.
> - Comparing conventional and pre-training methods is not fair because pre-training on large calculation datasets requires extensive computational costs. Furthermore, selecting appropriate pre-trained models imposes additional costs.
>
> We are currently conducting the experiments to measure the prediction accuracy of the pretraining models to resolve the reviewer’s concerns. We will upload the evaluation results to the author’s rebuttal as soon as the experiments are finished.
>
> ---
> **Q1:** Thank you for your suggestion. We will add notations for model parameters of individual neural networks to distinguish separate models.

---

> > ### Comment · Reviewer_bQLK · 2024-11-25
> >
> > Thank you to the authors for the response! The revised manuscript has improved the clarity and included more results, especially on the embedding analysis and the unsupervised learning results.
> >
> > I agree with the point raised by most reviewers, which is also the point in W1, that the contribution of this paper is to incorporate effectively the electronic information of substructures which can be acquired from database retrieval, and the development of diffusion process to effectively learn an embedding from it. While the electron-aware representation is not very physically meaningful, this paper provides a practical approach for getting better representations by utilizing substructure information. It is still a novel approach.
> >
> > Regarding the improvement of DELID vs DELID_qm, I agree with the authors that the accuracy is consistently better across datasets. But across almost all datasets the error ranges overlap. I can still see that the improvement is primarily due to that the s_0 can be seen as essentially learning an embedding on top of s_T and G_T, and the embedding has also been trained by the property labels y. Since the information is all from s_T, G_T, G and y, the self-supervised diffusion process learning provides one way to generate such embedding effectively. My intuition is that for DELID_qm, if the model size is increased to be the same, and a proper embedding layer is used for s_T and G_T, then I would still expect a comparable performance as DELID.
> >
> > As for the point of the pretraining experiment, I agree with the authors that from the computational cost perspective, this method has a valuable contribution.
> >
> > Overall I think the quality and the clarity of the paper is improved, and would like to increase my ratings.

---

> > > ### Author Response · Authors · 2024-11-25
> > >
> > > Thank you for your constructive feedback! We understand your concerns about the diffusion process on $s$. Based on the review comments, we will conduct further study on DELID and DELID_qm to clarify the reason for the performance improvements by DELID.

---

### Official Review · Reviewer_KLby · 2024-11-04

**Soundness:** 4
**Presentation:** 4
**Contribution:** 4
**Rating:** 8
**Confidence:** 3

**Summary:**

The authors propose a method for learning molecular representations using self-supervised diffusion processes. This method can learn an "electron-aware" representation without requiring ground truth electron properties of the complete molecule. This is achieved by leveraging the synergy between the diffusion process at the graph level and the electron-level information. Additionally, by decomposing the complete graph into subgraphs (molecular fragments), the reverse diffusion process can utilize subgraph information readily available in chemical databases. Lastly, the method showed superior performance compared to state-of-the-art methods in various downstream tasks.

**Strengths:**

- Originality: The idea of learning electron-aware molecular representation without needing ground truth electron properties is novel and significant. The method is also original in how it leverages the synergy between the diffusion process and electron-level information.

- Quality: The method is well designed, and the experiments are thorough.

- Clarity: The paper is well written and easy to follow.

- Significance: The method showed superior performance on various tasks and can inspire future research in the field.

**Weaknesses:**

- The authors provide details on the diffusion process, but the actual process of predicting the properties is not described in enough detail. How are the learned representations used to make final property predictions?
- In Table 1, the benchmark methods are mostly GNNs and XGB. It would be better to compare with more diverse methods, such as Fingerprint + MLP/SVR, or other methods representing diverse paradigms.

**Questions:**

- Is there a typo in Equation 1? Should it be t=1 or t-1 in the denominator?
- Regarding the construction of s_T: Some properties, such as energy, are extensive properties, so the quantity depends on molecule size. Retrieval using Tanimoto similarity does not consider molecular size. If the retrieved fragment and the actual fragment have different sizes, the property could differ significantly. Will this negatively impact model performance? Additionally, is there a list of the 19 features used to construct Q_k? Is the final $s$ just the 19 features for the complete molecule?

---

> ### Author Response · Authors · 2024-11-20
>
> We sincerely appreciate the reviewer’s constructive feedback on our paper!
>
> **W1:** Thank you for your comment. As we focused on explaining the self-supervised diffusion process, there was some lack of explaining the final prediction process. As shown in Eq. (3), the target variable is calculated by sum of two trainable functions for the input atom-level molecular graph G and the estimated electron-aware representation s. Eq. (3) follows the physical convention that the measured molecular properties contain the uncertainty derived from the electronic structure [1]. Please check Fig. 2a that describes the entire forward process of DELID.
>
> [1] Uncertainty quantification in chemical systems. 2009. Int. J. Numer. Methods Eng.
>
> ---
> **W2:** Thank you for your suggestion. However, XGB and GNNs are advanced models of the conventional SVR and MLP, respectively. For an experimental demonstration, we measured R2-scores of SVR and MLP on the benchmark datasets. As shown in the following table, XGB, GNN, and DELID outperformed SVR and MLP.
>
> |Method|Lipop|ESOL|ADMET|IGC50|LD50|LC50|LMC-H|CH-DC|CH-AC|
> |---|---|---|---|---|---|---|---|---|---|
> |SVR|0.539 (0.014)|0.660 (0.044)|0.719 (0.013)|0.646 (0.026)|0.405 (0.020)|0.488 (0.041)|0.475 (0.017)|N/R|N/R|
> |MLP|0.491 (0.022)|0.574 (0.039)|0.677 (0.021)|0.401 (0.055)|0.459 (0.018)|N/R|0.489 (0.032)|0.412 (0.028)|0.255 (0.074)|
> |XGB-Mor|0.531 (0.024)|0.659 (0.045)|0.717 (0.021)|0.621 (0.040)|0.390 (0.133)|0.497 (0.016)|0.505 (0.018)|N/R|N/R|
> AttFP|0.710 (0.021)|0.909 (0.018)|0.851 (0.027)|0.807 (0.013)|0.513 (0.016)|0.642 (0.079)|0.456 (0.031)|0.441 (0.099)|0.296 (0.370)|
> DELID|0.782 (0.013)|0.912 (0.014)|0.834 (0.042)|0.844 (0.006)|0.566 (0.024)|0.644 (0.068)|0.532 (0.048)|0.886 (0.035)|0.885 (0.023)|
>
> ---
> **Q1:** Thank you for careful reading our paper. That is a typo, and “t-1” should be corrected to “t=1”.
>
> ---
> **Q2:** We understand the reviewer’s concerns. Some incorrect retrieval results can negatively impact the model accuracy. To handle the noise from the retrieval results, we designed DELID to learn a probability distribution of s instead of a deterministic embedding function from G_0 to s, as shown in Eq. (6). Although the retrieval process based on Tanimoto similarity is efficient, more accurate and chemically valid retrieval process will have to be studied in future work of DELID.
>
> First, we need to correct a typo “19 features” to “15 features”. A list of the 15 features was not provided in the manuscript. However, in Table 2, we categorized these features into three sets and measured the prediction accuracy of DELID for different set of electron-level features to show the prediction capabilities of DELID for different choices of the electron-level features in the implementation of the retrieval process. Based on the reviewer’s comment, we will add a list of the 15 features in Appendix F. Please check the Table 5 in the uploaded rebuttal file.
>
> As mentioned in the Introduction section and the subsection 3.2.1, s is the electron-aware molecular representation generated from the 15 electron-level features of the decomposed substructures. Therefore, s is different to the reconstructed 15 electron-level features of the input molecule. We will add more clear descriptions about s.

---

> > ### Comment · Reviewer_KLby · 2024-11-20
> >
> > Thanks for your detailed response!
> >
> > Just for clarification,
> >
> > **W1**: What is the model architecture for the final prediction process?
> >
> > **Q2**: What is the dimension for $s$?

---

> ### Author Response · Authors · 2024-11-20
>
> Thank you for your reply.
>
> **W1:** The model architecture for the final prediction process is illustrated as the "Prediction model" in Fig. 2a. However, the detailed architecture of the prediction model was not presented in Fig. 2a because its implementation is not limited to a specific model. In our work, we used a graph neural network and a fully-connected neural network for the implementation of f(G) and psi(s;G) in Eq. (3), respectively.
>
> ---
> **Q2:** The dimensionality of $s$ is a hyperparameter of DELID. Since $s$ is entered into the final prediction layer to predict the one-dimensional molecular property $y$, we set $s = 32$ to compress the latent information calculated from the complex molecular structures and electron-level information.

---

> > ### Comment · Reviewer_KLby · 2024-11-24
> >
> > Thank you! All my questions were answered promptly.

---

### Official Review · Reviewer_QeAj · 2024-11-04

**Soundness:** 3
**Presentation:** 3
**Contribution:** 3
**Rating:** 6
**Confidence:** 4

**Summary:**

The paper introduces a method called DELID (Decomposition-supervised Electron-level Information Diffusion) for molecular representation learning. DELID utilizes molecular decomposition to generate substructures and matches them with a subset of the QM9 dataset to retrieve electron-level information. The method comprises two diffusion models: one estimates the original molecular graph from the decomposed subgraphs, and the other estimates the electron-level information from the electron-level features of these substructures. Finally, DELID employs these estimates to predict molecular properties. This approach allows DELID to learn electron-aware molecular representations and predict properties of complex molecules effectively, leveraging both structural and electronic information.

**Strengths:**

This paper introduces a self-supervised diffusion models method to represent molecules by estimating electron-level information in complex molecules. The proposed method is efficient as it does not require expensive quantum mechanical calculations to estimate electron-level information. DELID achieves state-of-the-art prediction accuracy on extensive real-world molecular datasets.

**Weaknesses:**

1. The performance of DELID is dependent on the external quantum mechanical calculation datasets, such as QM9. The distribution and the quality of such datasets may influence the ability of the models.
2. After breaking down the molecule into substructures, the electronic information of each part is calculated, and thus the interactions between the parts can be neglected. For example, the electronic shift of two neighboring functional groups. It is doubtful that the diffusion model is able to construct such interactions well.
3. The decomposition from complex to simple molecules is unique, but the process of denoising from simple molecules to generate the original molecule is not unique, e.g. isomers. This is not well avoided by the electronic information generated using the self-supervised method.

**Questions:**

1. Diid the authors use only the middle layer embedding of the diffusion model in their predictions or did they also use the final estimated electronic information?
2. The difference between DELIDqm and DELID in table3 makes me wonder if the electron level of the diffusion model is well constructed for electronic information of complex molecules. Can the authors provide the difference between the final estimated electronic information and the information obtained from the dft calculations?
3. What is the difference between the molecules generated by the diffusion model and the original molecules?
4. Can the embedding vectors generated by the diffusion model  distinguish different molecules? For example using t-sne to visualize the embedding of a certain dataset and distinguish different categories of molecules with molecular weights or whether they are aromatic molecules?
5. How to construct the molecular graph and how to combine several subgraphs to generate a graph? How to represent the electron information from the QM9?

---

> ### Author Response · Authors · 2024-11-20
>
> We sincerely appreciate the reviewer’s constructive feedback on our paper!
>
> **W1:** We agree with the reviewer that the performance of DELID is dependent on the choice of the external dataset. However, since DELID decomposes the input molecule into small substructures, the coverage of the calculation datasets is not a problem in prediction capabilities of DELID. Table 8 in Appendix J experimentally demonstrates our claim.
>
> ---
> **W2:** Thank you for bringing this to our attention; we unintentionally omitted the implementation details of the diffusion models in DELID. The decomposed substructures are finally converted to an attributed graph G_T through the decomposition and retrieval processes. The diffusion probability on G_T is parameterized by GNN, which can learn the interactions between nodes. Therefore, the interactions between the decomposed substructures and their electron-level information are fully reflected in the molecular representation learning, as each substructure is defined as a node in G_T. We will add more description of the diffusion process on G_T, as shown in lines 230-232 of the new manuscript uploaded during this rebuttal period.
>
> ---
> **W3:** The diffusion process in DELID aims to generate informative molecular representation for given prediction tasks rather than to reconstruct the original molecule. However, we agree with the reviewer that if the purpose of the diffusion process in DELID were to reconstruct the original molecule from its decomposed substructures, the issue of isomers would need to be carefully addressed, as the mapping between a set of substructures and the original molecule is not a one-to-one function.
>
> ---
> **Q1:** As described in Eqs. (3) and (8), we used the final estimated electronic information denoted by s or s_0. The middle layer embeddings were not used for the final prediction. As shown in Eq. (8), the middle layer embeddings were used as the self-supervised label to learn the diffusion process on the electron-level information.
>
> ---
> **Q2:** As explained in lines 043-048, molecular structure optimization with DFT on real-world complex molecules is not feasible due to its time complexity. For this reason, we cannot obtain the DFT-calculated features of the complex real-world molecules on the benchmark datasets. Therefore, a direct comparison between the DFT-calculated features and the electron-aware molecular representation of DELID is not possible.
>
> Instead of the direct comparison with DFT, we plotted the electron-aware representation s, which is the final outputs of the diffusion process on the electron-level information, and labelled them with the target molecular properties on the experimental datasets. We used t-SNE to project s to the 2D feature space, as shown in Appendix M of the uploaded rebuttal file. As shown in Appendix M, DELID generated the electron-aware latent embeddings highly related to the target molecular property without the complete electron-level information about the input molecules. This result shows that the diffusion model in DELID learned a way to describe the target molecular properties based on the electron-level features of the molecular sub-structures. Furthermore, the ablation study on DELID_qm and DELID empirically shows that the diffusion model in DELID generated more informative electron-aware latent embeddings.
>
> ---
> **Q3:** As described in lines 266-267 and Eq. (8), the diffusion process on the molecular structures was designed to generate intermediate embeddings to guide the diffusion process on the electron-level information. Although the diffusion model on the molecular structure is trained to reconstruct the input molecular structure by maximizing log p(G), the entire model is finally trained to maximize log p(y, s, G), as shown in Eq. (2). In other words, the diffusion model generates the molecular embeddings that describe both the input molecular structure and the target molecular property. Therefore, the original molecule and the output of the diffusion model are different in our problem setting to maximize p(y, s, G).

---

> > ### Comment · Reviewer_QeAj · 2024-11-20
> >
> > Thanks for your clarification. The embeddings visualization is great and shows the ability of your work. Using substructures as nodes makes sense to me.
> > As shown in Fig 2(a), the diffusion process on $G_t$ generates the original molecule and the diffusion process on $s_t$ generates the complete electron information. But your reply seems that the first one cannot get the original molecule and the complete electron information you mentioned does not mean the electron information for the whole complex molecule but the quantum noise. The DELID model aims to combine the structure information and electron information to get psi(s,G)  mentioned in Eq.(3) but reconstruct the molecules.
> > The f(G) in Eq.(3) is from G_0 which is generated by the diffusion process or the original G?

---

> ### Author Response · Authors · 2024-11-20
>
> **Q4:** We plotted the electron-aware representation s using t-SNE for further qualitative analysis, as shown in Appendix M in the uploaded rebuttal file. As shown in Fig. 5, DELID generated s highly related to the target molecular properties, even though s is essentially generated without the complete electron-level information about the input molecules.
> As shown in Appendix N in the uploaded rebuttal file, we plotted s for the molecular weight, which is an underlying variable determining several molecular properties. As shown in Fig. 6, DELID generated s that roughly describes the underlying molecular weight, even though the molecular weight was not provided for training DELID. However, the embedding results for the existence of the aromatic ring were not informative because the target molecular properties have very weak relationships with the existence of the aromatic ring. Please note that DELID essentially aims to generate electron-aware representation that describe the target molecular property, as shown in the objective function in Eq. (2).
>
> ---
> **Q5:**
>
> **Regarding the graph construction:** We followed a common molecular graph representation, as described in lines 30-33.
>
> **Regarding the combination of subgraphs:** As described in lines 227-230, the decomposed subgraphs are combined as a graph defined by a set of subgraphs, subgraph edges, and electron-level features.
>
> **Regarding the electron information from QM9:** We take 15 energy- and polarity-related electron-level features of the small molecules from the QM9 dataset, as described in lines 325-328. For each decomposed substructure, we assigned the 15 electron-level features by comparing the molecular similarity between the decomposed substructures and the small molecules. Section 3.2.3 presents the conceptual and mathematical descriptions of the information retrieval process on the external calculation datasets.

---

> ### Author Response · Authors · 2024-11-21
>
> Thank you for your reply. We apologize for some confusing notations and answers.
>
> **Clarification of the difference between $G$ and $G_0$:** We would like to clarify the difference between $G$ and $G_0$. $G$ is the original atom-level molecular graph, while $G_0$ is the latent embedding that follows the distribution of $G$ rather than directly reconstructed $G$ in our problem setting of Eq. (2).
>
> **Clarification of $G$ in $f(G)$:** The $G$ in $f(G)$ of Eq. (3) is the original atom-level molecular graph $G$. We have also tried $G_0$ instead of $G$ for $f(G)$, but we observed performance degradation. We believe the primary reason is that the diffusion process of obtaining $G_0$ from $G_T$, which represents the decomposed substructures of $G$, is not meaningful for small molecules $G$ that cannot be decomposed into chemically meaningful substructures. For this reason, to ensure the universality of DELID across molecules of varying sizes, we used $G$ instead of $G_0$ for $f(G)$.
>
> We thank the reviewer again for raising a good point of confusion. Accordingly, we entirely revised the descriptions related to $G_0$ and clarified the definition of $G_0$ as shown in lines 224-227. We also revised Fig. 2a. Please check the uploaded rebuttal file.

---

> > ### Comment · Reviewer_QeAj · 2024-11-25
> >
> > Thank you to the authors for their clarifications, which addressed my questions. While the DELID approach does not reconstruct molecular and electronic information—a limitation that raises concerns regarding the application of the diffusion model—it is clear that the paper is both inspiring and contributes meaningfully to the field. I will therefore increase my score.

---

> > > ### Author Response · Authors · 2024-11-26
> > >
> > > Thank you for your comprehensive feedback. We truly appreciate the reviewer’s constructive comments.

---

### Official Review · Reviewer_3yqb · 2024-11-04

**Soundness:** 3
**Presentation:** 3
**Contribution:** 3
**Rating:** 8
**Confidence:** 4

**Summary:**

This paper presents a new approach for electron-aware molecular representation learning. The key motivation is to introduce electron-level information into the learning process. The framework is formulated as a self-supervised diffusion process maximizing $log p(y,s,G)$, which is implemented as follows: (1) the input 2D molecular graph is decomposed based on extended functional groups (EFGs) to obtain a noised adjacency matrix $G_T$; (2) for each of the components, quantum-chemical properties are retrieved from a support set (a subset of QM9 in the experiments) as noised electron-level information $s_T$; (3) the model is then required to denoise the adjacency matrix and the electron-level information and predict the final property y simultaneously.

The proposed method is evaluated on several experimental molecular property prediction datasets, showing higher prediction accuracies than existing methods. The proposed method is also shown to have better generalization capabilities by an experiment simulating data-scarce scenarios. An ablation study on model components is provided. The authors also provide experimental results for different electron-level features, different external support datasets, execution time, etc.

I feel that the paper is of good quality overall, but there are still some points that need clarifications and discussions.

**Strengths:**

* The paper is well motivated. I agree with the authors that a molecular representation model should be electron-aware. The proposed method implements this without requiring ground-truth labels for whole molecules, which is novel and interesting.
* The paper is also easy to understand in general. I was able to follow the paper with most of my questions answered in the text.
* Probably thanks to the fast training/inference time, the evaluation is very thorough and comprehensive. The proposed method, DELID, not only shows higher prediction accuracies than existing methods, but also exhibits lower computation consumption and better generalization capability. The comparative analysis on different feature selections, different support datasets and different decomposing methods makes one believe the current design is the optimal.
* The provided code is clean and easy to follow, with dataset files directly included. This gives a good impression that the work can be easily reproduced using the provided code.

**Weaknesses:**

- Interpreting $s_T$ as electron-level information could be confusing. $s_T$ of shape $N_{comp} \times m$, which are pre-calculated properties for the decomposed substructures, actually seems more like substructure-level information rather than electron-level information. First, the only reason why the features can be vaguely regarded as containing information about electrons is that they are computed using quantum chemistry methods like DFT. However, these methods run on conformations, not on 2D graphs. The retrieval process for $s_T$ actually links a 2D substructure to a 3D conformation, and the latter could even be of a similar but different molecule. Second, the used features are mostly molecular-level observables aggregated from electron density. These two reasons make $s_T$ less physical and less suitable to be regarded as electron-level information.
- $s_t (t = 0, 1, ..., T - 1)$ is not only unknown, but also hardly defined. I think it can only be interpreted as latents, not as electron-level or substructure-level information. For example, the meaning of $s_0$, an $N_{comp} \times m$ tensor, is not clear given that the components are already connected in its corresponding graph $G_0$.
- The method is called self-supervised diffusion, while the current implementation seems to rely on the availability of $y$ .

**Questions:**

- Given the points mentioned in Weaknesses, it is then unlikely the improvements of the proposed method come from a more physical design. Could you provide an intuitive explanation about *why* do the improvements occur?
- A few important implementation details seem to be missing from the current version of paper. For example, what exact value does $T$ have?
- Is $y$ required for the model to learn good molecular representations? If it is required, do models trained on different datasets predict similar $s_0$?
- In section 4.4, the ablation study on DELID, could you also list the `#params` of the variants? What happens if the `#params` of the ablated variants match that of the complete DELID?

---

> ### Author Response · Authors · 2024-11-20
>
> We sincerely appreciate the reviewer’s constructive feedback on our paper!
>
> **W1:** Thank you for your comments from the physical perspective. Since we also agree that s_T is a feature aggregated from the electron density, not from the electronic structures directly, we named s as electron-aware representation instead of electron-level feature of G. If the word “electron-level information” is confusing, we can rename s_T as electron-derived information. However, it is still valid that DELID learned the molecular representation informed by electron-derived information beyond the conventional atom-level descriptors, and the ablation study in Table 3 experimentally demonstrates the effectiveness of our approach.
>
> As commented by the reviewer, different molecules can be matched to the decomposed substructures in the retrieval process of DELID, and it causes the noises in the input data. However, we designed the representation learning process based on the probabilistic models to handle the input noises from the mismatched information. As shown in Table 3, DELID achieved consistent accuracy improvements over DELID_qm by employing our probabilistic modeling based on the diffusion processes.
> For further analysis, we plotted s_0 using t-SNE and colored s_0 with the corresponding target molecular property. Please check Appendix M in the new manuscript uploaded in this rebuttal period. As shown in Appendix M, s_0 (= s) showed strong correlations with the target molecular properties for most experimental datasets, even though s_0 was essentially generated without the complete electron-level information about the input molecules.
>
> ---
> **W2:** We understand that the electron-aware molecular representation s_0 is somewhat ambiguous from physical perspective. However, as mentioned in Section 3.1, s_0 is learned to contain the uncertainty derived from the electronic density of the input molecule, and we devised the conditional diffusion process to learn the electron-derived uncertainty based on the DFT-calculated features of the decomposed substructures. To avoid the confusion as pointed out in the reviewer’s comment, we will rename s_t as “electron-aware latent embedding” instead of electron-level information. However, the importance of learning s_t was evident, as shown in the R2-scores of DELID_at and DELID in the ablation study. The experimental results of the ablation study demonstrated that considering the electron-derived uncertainty is important in building an accurate prediction model on experimental molecular datasets.
>
> ---
> **W3 & Q3:** We understand that some descriptions about the self-supervised learning of the conditional diffusion process is somewhat confusing. More precisely, the entire training process of DELID is supervised by the label data y. However, the sub-training process of the conditional diffusion process on s is conducted without the label data for s_0. For this reason, we referred to the conditional diffusion process on s as the self-supervised diffusion process.
>
> However, if we optimize the model for molecular embedding rather than molecular property prediction, the objective function will be given by log p(s,G), and consequently, the conditional diffusion model in DELID can be optimized without y. Eq. (8) mathematically shows that log p(s, G) can be maximized without the label data for y and s_0.
>
> The training process of the conditional diffusion model fundamentally does not require the label data for s_0, but s_0 can be affected by the provided molecular property y in our problem setting to maximize log p(y, s, G). Appendix M in the uploaded rebuttal file demonstrates that DELID generates informative s highly related to the target molecular properties, even though s is essentially generated without the complete electron-level information about the input molecules.
>
> ---
> **Q1:** Chemically, specific substructures can dominate the entire molecular properties, such as solubility, toxicity, and optical properties. DELID leverages both the atom- and electron-level information about the substructures, and it consequently provides more informative descriptors about the substructures without expensive quantum mechanical calculations. The accuracy improvements on the Lipop, IGC50, LC50, CH-DC, and CH-AC can be interpreted from this chemical perspective.
>
> ---
> **Q2:** Thank you for your careful review of the manuscript. The values of T were dependent on the benchmark datasets. We selected T within a range [3, 10] because the benchmark datasets are small. In addition to T, we will check missing implementation details and present the detailed configurations of the final implementations.

---

> > ### Comment · Reviewer_3yqb · 2024-11-21
> >
> > Thank you for the informative responses. My major concern still lies in the interpretation of the proposed method. In my view, interpreting it as "electron-aware self-supervised diffusion" is not precise and could be misleading. The reasons are as follows:
> > 1. $s_t (t=0,1,..,T-1)$, which is regarded as "electron-level information" by the authors, has no clear physical meaning. The authors have also acknowledged this point in the comments.
> > 2. The new results and responses do not fully resolve my concerns in this aspect. The new t-SNE results of `s_0` in the Appendix M mainly show that the learned `s_0` has strong correlations with the target molecular properties. The authors interpret this as an evidence of `s_0` being physical, but I would like to point out that this could also be interpreted as an evidence as `s_0` being nothing special but some latents. For another evidence, as is also mentioned in the authors' response to **W3 & Q3**, the target property `y` affects the learned `s_0`.
> >
> > In my view, the true contributions of this paper are: 1. the decomposing scheme; 2. the way to obtain and leverage features (`s_T`) of the decomposed molecule; 3. a novel network design that resembles diffusion. However, the paper describes the proposed method in a physical-looking way albeit it is actually not that physical.
> >
> > To sum up, I feel that the paper is novel and meaningful, but it is presented in an imprecise way. I would like to keep the current  gradings for the time being.

---

> ### Author Response · Authors · 2024-11-20
>
> **Q4:** The number of trainable parameters of DELID_at, DELID_et, DELID_qm, and DELID are about 610k, 97k, 650k, and 680k, respectively. We built the ablated variants with about 650k-690k model parameters and measured the R2-scores on the benchmark datasets. As shown in the following table, despite a slight accuracy improvement on DELID_et, the experimental results of the ablation study were not different from Table 3 in the manuscript. This result was not surprising because DELID_qm already had almost the same number of parameters as DELID.
>
> | Method | Lipop|ESOL|ADMET|IGC50|LD50|LC50|LMC-H|CH-DC|CH-AC|
> |---|---|---|---|---|---|---|---|---|---|
> |DELID_at|0.725 (0.016)|0.812 (0.036)|0.795 (0.027)|0.762 (0.025)|0.531 (0.026)|0.505 (0.112)| 0.458 (0.029)| 0.366 (0.028)|N/R|
> |DELID_et|0.315 (0.015)|0.507 (0.024)|0.612 (0.031)|0.523 (0.018)|0.287 (0.041)|0.388 (0.105)|0.270 (0.051)|0.632 (0.041)|0.605 (0.027)|
> |DELID_qm|0.776 (0.005)|0.906 (0.015)|0.823 (0.005)|0.829 (0.010)|0.530 (0.027)|0.622 (0.079)|0.505 (0.039)|0.843 (0.026)|0.878 (0.012)|
> |DELID|0.782 (0.013)|0.912 (0.014)|0.834 (0.042)|0.844 (0.006)|0.566 (0.024)|0.644 (0.068)|0.532 (0.048)|0.886 (0.035)|0.885 (0.024)|

---

> > ### Comment · Reviewer_3yqb · 2024-11-21
> >
> > Thank you for the additional results. Could you give a more detailed description for `DELID_qm`? In particular, what exactly is the function $\psi(G_T, s_T)$? Does it uses `A_0`?

---

> ### Author Response · Authors · 2024-11-23
>
> **Answer for the implementation of DELID_qm**
>
> We apologize for the confusion that we may have caused regarding formulation of DELID_qm in Section 4.4. $\psi(G_T, s_T)$ is a fully connected layer on a simple concatenation of the embeddings of $G_T$ and $s_T$, obtained by GNNs and MLPs, respectively. To be consistent with Eq. (3), we revised $\psi(G_T, s_T)$ to $ \psi(G_T, s_T;G)$.
>
> ---
> **Answer for s**
>
> Thank you for the clarification of the contributions. We agree with the viewpoints of the reviewer. Also, some misleading descriptions should be polished. However, we would like to kindly argue that our proposed method is indeed "electron-aware" due to the following two reasons.
>
> - Although there is no clear physical meaning in $s_t$ at $t = 0, 1, …, T-1$, we can ensure that they are essentially derived from the electron-related information, because the diffusion process on $s$ starts from the fragmented electron-derived features $s_T$ injected to $G_T$.
>
> - $s_0$ is generated only from the decomposed substructures and their electron-derived features. There is no complete information about the input molecule in the diffusion process of calculating $s_0$. Therefore, we can ensure that the diffusion process on $s$ learns some rule to assemble the fragmented physical information into the target variables of the complete chemical systems.
>
> ---
> **Answer for the self-supervised setting**
>
> **Regarding the self-supervised learning:** We agree that the diffusion process on $s$ can be affected by the target molecular property $y$, since the embedding networks that make up the diffusion processes are essentially trained with $y$. More precisely, the diffusion process on $s$ is trained with the self-supervised labels generated through the diffusion process on $G$ for the embedding networks optimized with $y$.
>
> **Regarding representation learning in unsupervised settings:** To address the reviewer’s concern, we trained the diffusion model on $s$ in a completely unsupervised setting to maximize $\log p(s, G)$ to show the representation capabilities of the diffusion model on $s$. As shown in Appendix P of the uploaded rebuttal file, the Wasserstein distance between the data distributions of the original molecular graph $G$ and the diffusion output $s$ consistently decreased as the diffusion model on $s$ was optimized. Please recognize the representation capabilities of DELID in the unsupervised setting.
>
> **Regarding the unsupervised DELID for the downstream prediction tasks:** Furthermore, we conducted additional experiments to evaluate the performance of DELID in the unsupervised setting through the following three steps. (1) We trained DELID to maximize $\log p(s, G)$. (2) We build a simple fully connected neural network that employs the diffusion output $s_0$ of the unsupervised DELID. (3) We measured the $R^2$-scores of the FCNN following the unsupervised DELID (denoted by NN-DELID). As shown in Appendix Q, NN-DELID outperformed the competitor methods for most benchmark datasets. However, the $R^2$-score of DELID was generally higher than that of NN-DELID. This experimental result demonstrates two conclusions as follows.
> - DELID works well even in the fully unsupervised setting, which aims to maximize $\log p(s, G)$.
> - Although the accuracy improvements are within the standard deviations in many cases, we can get further accuracy improvements by training DELID to maximize $\log p(y, s, G)$ instead of log $p(s, G)$ for the downstream prediction tasks.
>
> Based on the reviewer’s comments, we will revise to clarify the contributions of this work and to present the experimental results of DELID in the unsupervised setting. Thank you for your insightful comments.

---

> > ### Comment · Reviewer_3yqb · 2024-11-24
> >
> > Thank you for the new results. The new results for the fully self-supervised setting are vital for recognizing the contributions of this paper. Although there can still be an alternative interpretation of the results that the `s_0` are just continuous latents that captures the connection pattern distribution, such discussions could be less beneficial as the proposed method shows practicality to some extent. Also, I appreciate the authors' open-mindedness and devotion to make the contributions clearer. I think the current version of this paper qualifies to be published in ICLR and would like to increase my ratings. Congratulations!

---

> ### Author Response · Authors · 2024-11-24
>
> Thank you for your comprehensive feedback. We truly appreciate the reviewer’s recognition of the additional experimental results!

---

### Official Review · Reviewer_Kwdq · 2024-11-09

**Soundness:** 3
**Presentation:** 2
**Contribution:** 3
**Rating:** 6
**Confidence:** 4

**Summary:**

The paper proposes to enhance molecular representation learning by incorporating electron-level information via diffussion process, which eliminates the need of expensive quantum calculations. The proposed methods are evaluated on real-world experimental property prediction datasets to show the applicability and effectiveness, and also on simulation dataset such as QM9 for further discussion. Several analysis are conducted to assist the findings.

**Strengths:**

- The paper proposes a new perspective on enhancing molecular representation learning by leveraging the diffusion process.
- The proposed method save the computational costs on quantum calculations.

**Weaknesses:**

- The decomposed substructures serve as the noise, what is the motivation and reason of such implementation? Why not insert noise to the molecules? The authors have defined such, but there are not sufficient explanations of why doing so.

- The authors mention that FFSEC has low calculation accuracy, then why still use it to calculate coordinates? Why not use commonly used Rdkit to generate?

- How to define the substructure size? The paper mentions that EFG-based method is used for decomposition, however they also remove the large molecules in QM9 due to the redundancy?

- Why only $R^2$-scores and F1 scores are reported? The commonly used evaluation metrics like MAE/RMSE and AUC/accuracy are not reported. The reported results are also not very significant compared with baselines, such as ESOL, ADMET, LC50.

- How is the 5-fold leave-one-out cross-validation implemented? Is it leave one fold out? Why not follow the standard evaluation method? What about the split type? Is it scaffold or random split?

- Have the authors compared with methods that utilize both 2D and 3D graphs, like 3d infomax [1] and GraphMVP [2]?

- The proposed method seems not universal as it is mentioned that only under certain circumstances the method would work well, like relatively large molecules. I am also confused about the term of electron, it seems that the proposed method is based on substructures, why is it called electron-aware?

- The paper needs proofreading, i.e., some tense is not consistent. Some contents show conflict. For example, in the related work SchNet is introduced under the 2D-GNNs, while it is actually 3D GNN, where the authors also mention such in the Competitor Methods part.


[1] Stärk, Hannes, Dominique Beaini, Gabriele Corso, Prudencio Tossou, Christian Dallago, Stephan Günnemann, and Pietro Liò. "3d infomax improves gnns for molecular property prediction." In International Conference on Machine Learning, pp. 20479-20502. PMLR, 2022.

[2] Liu, Shengchao, Hanchen Wang, Weiyang Liu, Joan Lasenby, Hongyu Guo, and Jian Tang. "Pre-training Molecular Graph Representation with 3D Geometry." In International Conference on Learning Representations.

**Questions:**

See weaknesses.

---

> ### Author Response · Authors · 2024-11-20
>
> We sincerely appreciate the reviewer’s constructive feedback on our paper!
>
> **W1**: The noise drawn from an appropriate prior, rather than a random noise, is important in learning accurate diffusion models [1, 2]. We employed a set of decomposed substructures of the input molecule as the noise of the diffusion model due to the following two advantages.
> - A set of substructures decomposed by EFG can be regarded as incomplete information about the input molecular structure because the edges between the substructures are missing in a set of substructures. Hence, a set of substructures can be used as the noised input of the molecular graph, which is drawn from a chemically valid prior.
> - By decomposing the input molecular graph into the substructures, we can utilize the readily accessible electron-level information about the small molecules by serving it via decomposed substructures. This strategy enables DELID to learn electron-aware molecular representation without expensive quantum mechanical calculations on real-world complex molecules.
>
> [1] Preserve your own correlation: A noise prior for video diffusion models. 2023. CVPR.
>
> [2] Diffusion models as plug-and-play priors. 2022. NeurIPS.
>
> ---
> **W2:** We used the FFSEC method implemented in RDKit [3] to obtain the 3D structures of the molecules. As described in lines 144-148, accurate calculations on real-world molecules are not feasible due to their cubic or greater time complexities with respect to the number of electrons in the molecules [4]. In chemical science, FFSEC-based methods are commonly used to approximately optimize atomic coordinates in feasible computational costs.
>
> [3] https://www.rdkit.org/docs/source/rdkit.Chem.rdForceFieldHelpers.html
>
> [4] Computational complexity of interacting electrons and fundamental limitations of density functional theory. 2009. Nature Phys.
>
> ---
> **W3:** The sizes of the decomposed substructures are automatically defined by chemical knowledge in the EFG method. We overlooked highlighting this benefit of the EFG method and will provide a more detailed explanation of the characteristics and implementation details of the substructure decomposition process. Thank you for bringing this important point to our attention. Please check lines 255-257 of the uploaded rebuttal file (revised manuscript).
> As mentioned by the reviewer, we removed large molecules because they are redundant. Table 8 shows that the prediction accuracy of DELID is consistent with the sizes of the QM9 subsets.
>
> ---
> **W4**:
>
> **Evaluation Metrics:** In chemical science, the R2-score that is a scale-invariant accuracy measure is more common than MAE and RMSE because the target molecular properties are measured in completely different units. Nevertheless, we added MAEs of the competitor methods and DELID to the revised manuscript, and the evaluation results were consistent with Table 1. Please check Appendix O in the uploaded rebuttal file. Also, please note that executing some 3D GNNs takes a lot of computational time, so, currently, we have uploaded MAEs of the 2D GNNs. We will upload MAEs of the 3D GNNs as soon as the experiments are finished
>
> **Accuracy Improvements:** We can presume the reasons for the marginal improvements by DELID as follows.
> - The target molecular properties of the ESOL and ADMET datasets are directly related to the molecular size that can be obtained by the atomic structures rather than the electronic attributes of the molecules [5, 6]. For this reason, the accuracy improvements by DELID, which is designed to learn electron-aware molecular representations, were marginal.
> - AttFP is specially designed to process drug-like molecules. We suppose that the domain knowledge implemented in AttFP would have helped improve the generalization capability of the prediction model on the very small LC50 dataset containing drug-like molecules.
>
> [5] Combinatorial chemistry & high throughput screening. 2011.
>
> [6] Scale-aware graph-based machine learning for accurate molecular property prediction. 2020. IEEE BigData.
>
> ---
> **W5:** We split the entire dataset into five folds. The four folds were merged and used as the training dataset, and the remaining fold was used as the test dataset. The training and test processes were repeated five times. We reported the mean and standard deviation of the evaluation metrics. This evaluation protocol (with or without the validation set) is usual in cheminformatics [7, 8]. We will correct the misleading word “5-fold leave-one-out cross-validation” to “5-fold cross-validation”.
> We used the random split to evaluate the generalization capabilities of the prediction models rather than their extrapolation capabilities. Chemically, the test datasets generated by the scaffold split mean the extrapolation sets.
>
> [7] Machine learning methods in chemoinformatics. Wiley Interdiscip. Rev. Comput. Mol. Sci.
>
> [8] MoleculeNet: a benchmark for molecular machine learning. 2018. Chem. Sci.

---

> ### Author Response · Authors · 2024-11-20
>
> **W6:** Please note that PhysChem is a model leveraging both 2D and 3D molecular structures. In addition to PhysChem, we implemented several 2D and 3D methods, but failed to evaluate them due to infeasible computational costs on real-world large molecules. As reported in Table 9, the 2D and 3D methods generally required 178 times more computation time. We are currently conducting the experiments to measure the prediction accuracy of 3D Informax, GraphMVP, and other methods. We will upload the evaluation results to the author’s rebuttal as soon as the experiments are finished.
>
> ---
> **W7:**
>
> **Regarding the universality of DELID:** We agree with the reviewer’s concern regarding the universality of our proposed method. To ensure the performance and applicability of DELID on small molecules, we designed DELID to leverage both the atom- and electron-level descriptors in molecular representation learning, as shown in Eq. (3). Since the latent representations generated from the atom-level descriptors are independent on the diffusion process, DELID can still predict the target molecular properties by leveraging the latent representations from the atom-level descriptors, even if the decomposition process cannot produce meaningful substructures. Therefore, DELID is universally applicable regardless of the molecular size. In addition to the mathematical modeling, we experimentally verified that DELID still achieved the highest prediction accuracy on the IGC50 and LC50 datasets, which contain relatively small molecules as presented in Table 4.
>
> **Regarding the term of electron:** As described in Eq. (9), the substructure is used as a container for carrying the electron-level information s_{qm,i}, and a set of the substructures with s_{qm,i} is used as the input data of DELID. Based on the substructure decomposition and information retrieval processes, DELID can learn latent molecular representations based on the electron-level information s_{qm,i} beyond the conventional atom-level descriptors without expansive quantum mechanical calculations.
>
> ---
> **W8:**
> Thank you for your careful comments. We will do proofreading more carefully to correct incorrect descriptions in the manuscript.

---

> > ### Comment · Reviewer_Kwdq · 2024-11-25
> >
> > Thank the authors for the detailed explanations, which address most of my concerns. Although I still hold concerns about the electron term as this method does not seem actually incorporate electron from the physical perspective, the overall method introduces merits. I will raise my score.

---

> > > ### Author Response · Authors · 2024-11-25
> > >
> > > We sincerely appreciate the reviewer’s constructive feedback. We will clarify the contributions of our work based on the review comments.

---

### Comment · Area_Chair_MPG1 · 2024-11-22

Reviewers,

The authors have posted their rebuttals. Could you please check their responses and engage in the discussions? Please also indicate if/how their responses change your opinions.

Thanks,

AC

---

### Meta-Review · Area_Chair_MPG1 · 2024-12-19

**Metareview:**

This paper proposes to incorporate electron-level information for molecular learning. To eliminate the need of complete ground truth electron properties of a molecule, the problem is formulated as a self-supervised diffusion process with several novel techniques introduced. All the reviewers have positive opinions toward the paper, and two of them are excited about the paper.

**Additional Comments On Reviewer Discussion:**

Originally, the reviewers raised some concerns, most of which have been successfully addressed during the rebuttal and discussions. Thus, I recommend the acceptance of the paper. I encourage the authors to tackle the remaining concerns from reviewers in the final version, such as clearly showing how to incorporate electron information from the physical perspective, etc.

---

### Decision · Program_Chairs · 2025-01-22

Accept (Poster)